



# Response of middle atmospheric temperature to the solar 27-day cycle: an analysis of 13 years of MLS data

Piao Rong[1,2,3], Christian von Savigny[3], Chunmin Zhang[1,2], Christoph G. Hoffmann[3], and Michael J. Schwartz[4]

[1]School of Science, Xi'an Jiaotong University, No.28, Xianning West Road, 710049 Xi'an, China
[2]Institute of Space Optics, Xi'an Jiaotong University, No.28, Xianning West Road, 710049 Xi'an, China
[3]Institute of Physics, University of Greifswald, Felix-Hausdorff-Str. 6, 17489 Greifswald, Germany
[4]Jet Propulsion Laboratory, California Institute of Technology, Pasadena, 91109 CA, USA

**Correspondence:** Chunmin Zhang (zcm@xjtu.edu.cn)

**Abstract.** This work focuses on studying the presence and characteristics of solar 27-day signatures in middle atmospheric temperature observed by the Microwave Limb Sounder (MLS) on NASA's Aura spacecraft. The 27-day signatures in temperature are extracted using the superposed epoch analysis (SEA) technique. We use time-lagged linear regression (sensitivity analysis) and a Monte-Carlo test method (significance test) to explore the dependence of the results on latitude and altitude, on
solar activity and season, as well as on different parameters (e.g., smoothing filter, window width and epoch centers). Using different parameters does impact the results to a certain degree, but it does not affect the overall results. Analyzing the 13-year data set shows that highly significant solar 27-day signatures in middle atmospheric temperature are present at many altitudes and latitudes. A tendency to higher temperature sensitivity to solar forcing in the winter hemisphere is found. In addition, the sensitivity of temperature to solar 27-day forcing tends to be larger at high latitudes than at low latitudes. For solar 11-year
minimum conditions no statistically significant identification of a solar 27-day signature is possible at most altitudes and latitudes. Several results we obtained suggest that processes other than solar variability drive atmospheric temperature variability at periods around 27-days. Comparisons of the obtained sensitivity values with earlier experimental and model studies show good overall agreement.

## 1  Introduction

The 27-day solar cycle is caused by the sun's differential rotation, which leads to apparent variations in solar flux with a period of about 27 days (e.g., Sakurai (1980) and references therein). Previous studies have identified solar 27-day signatures in many different atmospheric parameters, e.g., noctilucent clouds (e.g., Robert et al., 2010), mesospheric water vapor (e.g., Thomas et al., 2015), tropical upper stratospheric ozone (e.g., Hood, 1986; Fioletov, 2009), the middle atmospheric odd hydrogen species

(e.g., Wang et al., 2015), upper mesospheric atomic oxygen (Lednyts' kyy et al., 2017), and especially in temperature (e.g.,





Hood, 1986; Keating et al., 1987; Hood et al., 1991; Hall et al., 2006; Dyrland and Sigernes, 2007; Robert et al., 2010; von Savigny et al., 2012; Thomas et al., 2015; Hood, 2016) in the middle atmosphere. The term "middle atmosphere" refers to the height region of approximately 15 – 90 km and comprises the stratosphere and mesosphere. While a significant number of experimental studies investigated solar-driven 27-day variations in stratospheric and mesospheric parameters, the physical/chemical

mechanisms leading to these signatures are, in many cases, not well understood. Therefore, it has become a highly interesting subject to study atmospheric variations due to the 27-day solar activity cycle in middle atmospheric parameters.

First, we briefly outline the existing experimental and modelling studies on 27-day solar periodicities in temperature of the middle atmospheric region.

Ebel et al. (1986) reported observations of solar-driven temperature deviations of about 1.5 K at 80 km in the tropics and

argue that since the response to solar activity (27-day and 13-day) is mainly determined by the dynamical properties of the middle atmosphere, the strongest perturbations should occur at middle and higher latitudes. The analysis covers the years from 1975 to 1978 and is based on temperature measurements with the Nimbus 6 Pressure Modulator Radiometer (PMR). Keating et al. (1987) also identified a 27-day signal in tropical mesospheric temperature (50 – 70 km) in the 1980s using Nimbus 7 Stratosphere and Mesosphere Sounder (SAMS) temperature data, and found a maximum sensitivity at 70 km. Hood (1986) and

Hood et al. (1991) used Nimbus-7/SAMS (Stratosphere and Mesosphere Sounder) measurements at low latitudes to determine the temperature sensitivity to solar forcing at the 27-day scale for altitudes extending up to about 90 km. Brasseur (1993) used a two-dimensional chemical-dynamical-radiative model of the middle atmosphere to investigate the potential changes of temperature in response to the 27-day variation in the solar ultraviolet flux. The temperature response to solar variability has not been considered at altitudes above 60 km in Brasseur (1993), because several radiative processes specific to the mesosphere

had not been treated in detail. Zhu et al. (2003) investigated the ozone and temperature responses in the upper stratosphere and mesosphere through analytic formulations and the Johns Hopkins University Applied Physics Laboratory (JHU/APL) 2D chemical–dynamical coupled model, showing an increasing sensitivity of temperature to the solar UV forcing with increasing latitude and altitude. Hall et al. (2006) and Dyrland and Sigernes (2007) identified signatures with periods near 27 days in winter time meteor radar temperature time series at 90 km and for latitudes of 70° N and 78° N. Gruzdev et al. (2009)

analyzed the effects of the solar rotational (27-day) irradiance variations on the chemical composition and temperature of the middle atmosphere as simulated by the three-dimensional chemistry-climate model HAMMONIA. They found that the response sensitivities of temperature to solar activity generally decrease when the forcing increases and in the extra-tropics the response was found to be seasonally dependent with typically higher sensitivities in winter than in summer. Robert et al. (2010) identified a solar-driven 27-day signature in mesospheric temperatures at middle and high latitudes during hemispheric

summer applying a cross-correlation analysis on MLS/Aura measurements. von Savigny et al. (2012) reported on a 27-day signature in equatorial mesopause (87 km) temperatures derived from Envisat/SCIAMACHY (SCanning Imaging Absorption spectroMeter for Atmospheric CHartographY) observations of the OH(3–1) Meinel band in the terrestrial nightglow. Thomas et al. (2015) investigated solar-driven 27-day variations in temperature profiles in the high-latitude summertime region for altitudes between 70 and 90 km and observed with the Solar Occultation for Ice Experiment (SOFIE) on the Aeronomy of Ice

in the Mesosphere (AIM) satellite. Hood (2016) analyzed daily ERA-Interim reanalysis data for three separate solar maximum





periods and confirmed the existence of a temperature response to 27-day solar ultraviolet variations at tropical latitudes in the lower stratosphere (15 – 30 km).

Besides, an influence of 27-day variability on tropospheric parameters is also debated (e.g., Hoffmann and von Savigny (2019) and references therein). Our paper focuses on the discussion of the middle atmosphere, so we do not discuss many details related to tropospheric parameters here.

In brief, previous studies found that variations of solar spectral irradiance at the 27-day time scale affect atmospheric temperature based on different observational and modelling data sets. However, for many atmospheric species and parameters, the processes leading to the observed 27-day signatures are not well understood. In addition, the statistical significance of the identified signatures is often difficult to establish.

This paper investigates the presence and characteristics of solar 27-day signatures in middle atmosphere temperature observed by the Microwave Limb Sounder (MLS). The MLS data set is uniquely suited for this purpose, because it provides global daily coverage and covers more than a 11-year solar cycle. We employ the solar Mg II index as the solar proxy. In this study, the superposed epoch analysis (SEA), time-lagged linear regression (sensitivity analysis), and a Monte-Carlo test method (significance test) are used. To investigate how robust the results are, the dependence of the results on parameters of the analysis methods (e.g., smoothing filter, window width and epoch centers), on the time of measurement (e.g., temperature observation time, solar activity and season), and on latitude and altitude are investigated.

The remainder of the paper is organized as follows: Section 2 describes the MLS temperature data set and the Mg II index data used in this study; Section 3 describes the analysis process and the main features of the SEA, the sensitivity analysis and the significance test; In section 4 the analysis results are presented, discussed and compared to earlier studies; Conclusions are provided at the end.

## 2 Data sets

### 2.1 Mg II Index

The core-to-wing ratio of the Mg II doublet (280 nm) in the solar irradiance spectrum, i.e., Mg II index, is frequently used as a proxy for tracking solar activity from the ultraviolet (UV) to the extreme ultraviolet (EUV) associated with the 11-year solar cycle (22-year magnetic cycle) and solar rotation 27-day cycle (Cebula and Deland, 1998; Dudok de Wit et al., 2009).

The Mg II index is a dimensionless proxy, but the relationship between the Mg II index and other solar proxies, e.g., the Lyman-$\alpha$ or the F10.7 cm radio flux or can be easily established by a linear regression (e.g., von Savigny et al., 2012, 2019). The F10.7 cm radio flux is usually given in solar flux units (sfu), which are equal to $10^{-22}$ W m$^{-2}$ Hz$^{-1}$. So, the results can be compared well with other research results.

For this study we employ the Bremen daily Mg II index composite data as the solar proxy, which is available from 1978 to present and is derived from four data sets, i.e., the Global Ozone Monitoring Experiment (GOME), SCIAMACHY, GOME-2A, and GOME-2B. The most recent information on the Mg II data can be found in Snow et al. (2014). The top panel of Figure 1 shows the Mg II index data from 2005 to 2017 which is used in this analysis.



## 2.2 MLS on Aura

The National Aeronautics and Space Administration (NASA) Earth Observation Satellite Aura has been in a near-polar 705 km altitude orbit since 2004. The Microwave Limb Sounder (MLS) on Aura consists of seven radiometers observing emission in the 118 GHz, 190 GHz, 240 GHz, 640 GHz and 2.5 THz regions. The MLS measurements provide vertical profiles of
temperature, geopotential height, several atmospheric trace species and ice water content of clouds with near-global coverage on a daily basis (Waters et al., 2006; Livesey et al., 2018).

MLS temperature is retrieved primarily from MLS measurements of the thermal emission of $O_2$ near 118 GHz and 240 GHz (Schwartz et al., 2008). The isotopic 240 GHz line is the primary source of temperature information in the troposphere (extending the profile down to about 9 km), while the 118 GHz line is the primary source of temperature information in the
stratosphere and above (from 90 km down to about 16 km) (Livesey et al., 2018).

In this work, we use the MLS Level 2 temperature product version 4.2. MLS version 4.2 temperature is available from 2 August 2004 to present. The precision and accuracy of the MLS temperature data product are shown in Table 3.22.1 of Livesey et al. (2018). The precision is 1 K or better in the troposphere and lower stratosphere (from 261 hPa to 3.16 hPa), degrading to 3.6 K in the upper mesosphere (at 0.001 hPa). The observed biases based upon comparisons with analyses and other previously
validated satellite-based measurements range from $-2.5$ K to $+1$ K in the troposphere and lower stratosphere, increasing to $-9$ K at the highest altitude. The recommended useful vertical range for scientific studies is between 261 hPa (10 km) and 0.001 hPa (96 km), and the vertical resolution varies between 3.6 km (at 31.6 hPa) and $13 - 14$ km (at 0.001 hPa). The horizontal resolution is $\sim$165 km between 261 hPa and 0.1 hPa and degrades to 280 km at 0.001 hPa. To investigate the presence of a 27-day solar cycle signature in the temperature data set and to keep the annual data complete, the period from January 1, 2005
to December 31, 2017 was selected as shown in the bottom panel of Figure 1. In the following analysis, we first employ the day and night averaged MLS temperature data. In section 4.1.1 and 4.2.1 we investigate how the results change if daytime (or nighttime) measurements only are employed for the analysis.

## 3   Methodology

The approach employed to analyze the 27-day solar cycle signal in temperature is illustrated in Figure 2. First, temperature
and Mg II index anomalies are calculated (see section 3.1). Next, the SEA method is applied to the temperature and Mg II index anomalies to obtain the epoch-averaged temperature and Mg II index anomalies (section 3.2). Then, the epoch-averaged temperature and Mg II index anomalies are used to perform the sensitivity analysis (section 3.3) and the significance test (section 3.4). The individual steps are described in detail in the corresponding subsections.

In the process, different input observational and statistical parameters may affect the results. For example, the results may
depend on whether daytime, nighttime or daily averaged MLS temperature data is used for the analysis. Other parameters that may affect the results are latitude and altitude, the width of the window used in the data pre-processing, the choice of the epoch centers (maxima or minima of Mg II index anomalies) applied for the SEA, the smoothing filter used to choose the maxima or





minima as epoch centers. In addition, the dependence of the results on solar activity and season also needs to be discussed. To check how these parameters affect the results, different tests are performed and described in section 4.

### 3.1  Data pre-processing

We defined a standard altitude grid with 36 levels from 20 to 90 km with a step size of 2 km and a standard latitude grid with 18 bins from 90° S to 90° N with a step size of 10°. MLS geopotential height was converted to geometric height using the height and latitude dependent formula provided by Roedel and Wagner (2011). The temperature data were averaged daily and zonally for each altitude and latitude bin between 1 January 2005 and 31 December 2017.

The bottom panel of Figure 1 shows the daily averaged temperature data for an altitude of 88 km and a latitude of 5° N (averaged zonally and over the 0 – 10° N latitude range). There are 5 data gaps and 6 abnormal peaks. The data gaps occur in the following periods: days 453 – 458 (6-day gap in 2006), day 555 (1-day gap in 2006), days 2276 – 2298 (23-day gap in 2011), days 2605 – 2609 (5-day gap in 2012), and days 2630 – 2635 (6-day gap in 2012). Days are counted starting with January 1, 2005. These gaps exist in the observations at all latitudes and altitudes (see Figure 3). The white lines in Figure 3 indicate that temperature data is missing. The outliers/abnormal peaks visible in the bottom panel of Figure 1 occur on days 341, 417, 452, 1532, 1759, 3717. Note that the outliers appear on different days for different altitudes and latitudes. In order to investigate the presence of a 27-day solar cycle signature in the temperature data set, it is necessary to avoid the invalid points (temperature gaps and outliers) in the SEA. This can be easily implemented in the SEA by ignoring the data gaps and outliers in the averaging procedure (see below).

Next, we apply a 35-day running mean and then calculate the anomalies as the deviation from the running mean for MLS temperature and the Mg II index time series. The resulting temperature anomalies for an altitude of 88 km and a latitude of 5° N are shown in the top panel of Figure 4. We define outliers as data points for which the absolute of the temperature anomaly exceeds 4 times the standard deviation of the anomaly time series. The bottom panel of Figure 4 shows the temperature anomaly with removed outliers. The width of the smoothing window is chosen as 35 days to remove the seasonal modulation of the temperature signal while leaving the variation at shorter time scales unaltered. In sections 4.1.1 and 4.2.1 we investigate how the results change if different window widths (e.g., 27 days and 50 days) are employed for the analysis. Those steps above are a preparation for the following SEA, the significance testing and sensitivity analysis.

### 3.2  Superposed epoch analysis (SEA)

To identify weak solar 27-day signatures in temperature time series affected by variability from various sources, the superposed epoch analysis method (SEA) (e.g., Howard, 1833; Chree, 1912) is an effective choice. The SEA is applied to the time series covering the period from January 2005 to December 2017.

An overview of the SEA is shown in Figure 5. First, the epoch centers need to be chosen. The local maxima in the Mg II index time series – reflecting maxima in solar spectral irradiance – can be used as the epoch centers (represented as Max 1 to Max N in Figure 5). The Mg II index maxima are identified in the un-smoothed (0-day) or 7-day or 13-day smoothed Mg II index anomalies as shown in Figure 6. The yellow, blue and red points represent the local maxima identified for the 0-day,





7-day and 13-day smoothed Mg II index anomalies, respectively. We discuss the impact of the smoothing filter on the results in sections 4.1.1 and 4.2.1. A similar method can be applied to choose the minima in the Mg II index times series and we compare the variation on the results by utilizing the maxima or minima of Mg II index anomalies for the SEA in sections 4.1.1 and 4.2.1.

Second, we choose 61 days centered at these solar maxima dates as an analysis epoch (i.e., 30 days before and after these maxima). The whole time series from 1 January 2015 to 31 December 2017 will be divided into $N$ epochs, and each epoch covers 61 days. Finally, the epoch-averaged temperature anomaly ($\mathrm{T}_{anomaly}[x]$) is obtained by averaging $N$ temperatures ($T_{epoch}^{x}$) of the corresponding day ($x$) in each 61-day epoch, see Equation (1).

$$\mathrm{T}_{anomaly}[x] = \frac{1}{N} \sum_{epoch=1}^{N} T_{epoch}^{x} \tag{1}$$

Here, $x$ represents an integer between -30 and 30. Similarly, the epoch-averaged Mg II anomaly is determined this way. Figure 7 (a) displays an example of the resulting epoch-averaged temperature (at 88 km and 5° N) and Mg II index anomalies. The Mg II index anomaly exhibits very symmetric behavior with a maximum at zero day time lag and minima near ±13 days, as expected. The epoch-averaged temperature anomaly also shows a clear maximum but with a time lag of 2 days, indicating that the response in mesospheric temperature to the solar forcing occurs with a certain time lag.

**3.3   Sensitivity analysis**

The resulting epoch-averaged temperature and Mg II index anomalies are used to determine the sensitivity of middle atmospheric temperature to changes in the solar activity represented here by the Mg II index. The relationship between temperature anomaly ($\mathrm{T}_{anomaly}[x]$) and Mg II index anomaly ($\mathrm{MgII}_{anomaly}[x]$) can be represented by a linear regression line (see Equation (2)) if the maxima in the epoch-averaged anomalies occur at the same time lag. The sensitivity is directly determined by

the slope ($k$) of a linear regression line to the data points, i.e., epoch-averaged temperature and Mg II index anomalies.

$$\mathrm{T}_{anomaly}[x] = b + k \times \mathrm{MgII}_{anomaly}[x] \tag{2}$$

However, as shown in Figure 7 (a), there is a time lag or shift ($l$) between solar maximum and temperature maximum. If the times of the maxima do not coincide, then an ellipse is fitted instead of a straight line. To remove the phase shift between the two anomalies, we need to shift the temperature curve by $l$ days to obtain the time-lagged epoch-averaged temperature

anomalies $\mathrm{T}_{anomaly}[x+l]$. Then the sensitivity parameter ($k$) is derived from Equation (3).

$$\mathrm{T}_{anomaly}[x+l] = b + k \times \mathrm{MgII}_{anomaly}[x] \tag{3}$$

The phase lag ($l$) can be determined by time-lagged cross correlation as shown in Figure 7 (b). The sensitivity for this particular combination of altitude and latitude is obtained by shifting the epoch-averaged temperature anomaly backwards by 2 days, i.e., $l = -2$, see Figure 7 (c). The sensitivity obtained for a 35-day window width, a 7-day smoothing filter and using

maxima of the Mg II index anomaly as the epoch centers is 190 ($\pm$ 15) K (Mg II index unit)$^{-1}$. The relationship between the





Mg II index and the F10.7 cm radio flux was established by a linear regression to annually averaged values for the years 2003 to 2010 ($\Delta$MgII / $\Delta$F10.7 = 0.0135 Mg II index unit (100 sfu)$^{-1}$) and the sensitivity value translates to 2.57 ($\pm$ 0.20) K (100 sfu)$^{-1}$. The result is in very good agreement with the conclusion of von Savigny et al. (2012). They analyzed zonally averaged OH(3-1) rotational temperatures at 87 km for the [0°, 20° N] latitude range using the Mg II index derived from SCIAMACHY

and found a temperature sensitivity to solar forcing in terms of the 27-day solar cycle of 182 ($\pm$ 69) K (Mg II index unit)$^{-1}$ or 2.46 ($\pm$ 0.93) K (100 sfu)$^{-1}$. We need to point out, however, that von Savigny et al. (2012) analyzed a much more limited time period – i.e., from April 2005 to October 2006 – compared to the results presented here. More comparisons of our sensitivity results to previously published ones are presented in section 4.2.2 and 4.2.3.

### 3.4 Significance testing

We use a similar Monte-Carlo test method as is used in von Savigny et al. (2019) to examine the significance of the obtained results. Instead of using local solar maxima as the epoch centers in the SEA, the epoch centers are chosen randomly and the SEA is repeated. The number of random epochs is the same as in the actual SEA. This procedure is carried out 1000 times. Then a sinusoidal function is used to fit every single random realization of the epoch averaged temperature anomaly. Comparing the amplitude of the fitted sinusoidal function of the 1000 random cases to the amplitude of the actual case, the

statistical significance of the SEA results can be evaluated. The amplitude and phase of fitted sinusoidal functions, as well as the fraction of random realizations with amplitudes larger than actual data are the results of the significance test. If the fraction of random realizations with amplitudes larger than the amplitude of the actual SEA is close to zero, then the 27-day signature in MLS temperature data is likely not a spurious signature. Figure 8 shows the results of the Monte-Carlo significance test at 88 km and 5° N. The local solar maxima used here are determined based on the 7-day smoothed Mg II index anomalies, which

were obtained by subtracting a 35-day running mean from the daily Mg II index data.

### 4 Results and discussion

The main purpose of the present work is to investigate the presence and characteristics of solar 27-day signatures in the middle atmosphere temperature observed by MLS. In order to investigate how robust the results are, different tests were performed, i.e., a significance test, a sensitivity test, and an investigation of the dependence of the results on real geophysical parameters

(i.e., solar activity, season, latitude and altitude) and on statistical/numerical parameters (i.e., window width, epoch centers, and smoothing filter).

### 4.1 Significance test results

The significance testing method was described in section 3.4. To investigate the dependence of the significance results on altitude and latitude, on the width of the window, on epoch centers and on the temperature observations, these tests were

performed at each altitude and latitude, for different window widths of 27 days, 35 days and 50 days, as well as different local





maxima chosen by 0-day/7-day/13-day smoothed temperature anomalies based on daytime, nighttime, and daily averaged temperature observations.

### 4.1.1 Dependence of the results on statistical parameters

The dependence of the results on the different parameters is carried out based upon temperature data in the tropical (5° N)
mesopause region (88 km). Table 1 lists the results for the different statistical parameters considered and for the different observational temperature (daytime, nighttime and daily averaged temperature) data sets. The maximum and minimum of the fraction of random realizations with amplitudes larger than actual data are underlined. The max-to-min variation of the fraction for the daytime temperature case is larger than the one for the nighttime and daily averaged temperature cases. In terms of daily averaged temperature, the maximum and minimum fractions are about 1.0 % and 0.0 %, respectively. That is, the variation of
the fraction is about 1.0 % for different input parameters. For nighttime temperature, the maximum and minimum fractions are about 1.9 % and 0.0 %, respectively. The max-to-min variation of the fraction is about 1.9 %, but for the daytime temperature, the maximum and minimum fractions are about 28.6 % and 1.5 %, respectively. The max-to-min variation of the fraction increases to about 27.1 %. The exact origin of this different behaviour of the daytime temperature data is currently unknown. More discussion on the dependence of the results on statistical parameters at different latitudes and altitudes will be given in
subsection 4.1.2.

### 4.1.2 Dependence of the results on latitude

We performed the significance test for the daily averaged temperature from 2005 to 2017 for the latitude range from 85° S to 85° N and the altitude range from 20 to 90 km. The resulting fraction of random realizations with amplitudes larger than the actual SEA is displayed in Figure 9 as a function of latitude and altitude. For the results shown in Figure 9 (a), the local solar
maxima used in the SEA are chosen from the 7-day smoothed temperature anomalies obtained by subtracting a 35-day running mean from the daily averaged temperature data. As shown in the Figure, there exists a complex pattern of latitude/altitude regions with low fractions indicating that the identified 27-day signatures are most likely not caused spuriously – making a solar origin likely. As shown in the Figure, low fractions of less than 10 % appear in the tropics for the altitude range of 40 – 60 km and 80 – 90 km, as well as at 40° N for the altitude of about 65 km. The low fractions (< 10 %) also appear at the high
latitudes, e.g., at 70 – 85° S for the altitude ranges of 30 – 40 km and 60 – 80 km, and at 80 – 85° N for altitudes of around 40 km.

    In addition, Figure 10 provides two examples of high and low significance cases. Figure 10 (a) shows the epoch-averaged Mg II index and temperature anomalies and the sinusoidal fit to the 3-day smoothed epoch averaged temperature anomalies for the actual SEA and for 1000 randomly chosen epoch ensembles at 88 km for a latitude of 5° N. There is no random sinusoidal
fit amplitude larger than the actual one, that is, the fraction of the significance test is 0.0 %. Figure 10 (b) is a significance test result for an altitude of 50 km and a latitude of 85° N. In this case 95.0 % of the random sinusoidal fit amplitudes are larger than the amplitude of the actual analysis.





In order to check the influence of the input parameters on the results at different latitudes, we show in Figure 9 the significance results for some of the combinations of input parameters yielding the largest fractions of random realizations with amplitudes larger than the actual SEA (see Table 1). The results obtained using a 27-day window width and 0-day smoothing filter are shown in Figure 9 (b). The results obtained using a 27-day window width, a 0-day smoothing filter, and daytime temperature data are shown in Figure 9 (c). The results obtained using a 50-day window width, a 0-day smoothing filter, nighttime temperature data, and minima of Mg II index anomaly are shown in Figure 9 (d). The low fraction regions obviously become smaller in 9 (b–d), but the locations of these regions have not changed. That means, different input parameters have an impact on the results, but will not affect the overall characteristics.

### 4.1.3 Dependence of the results on season

To determine whether the solar 27-day cycle signal in middle atmospheric temperature depends on season, the SEA and the subsequent significance tests were performed for winter and summer separately. We assume that "winter" includes the six months of October, November, December, January, February and March, and "summer" includes the other six months for the northern hemisphere. For the southern hemisphere, it is the opposite. More than three months for each season are considered here in order to increase the number of epochs available for analysis.

The significance testing results depending on season are shown in Figure 11 (a – b). The input parameters used in this analysis are the same as in the Figure 9 (a). In the southern hemisphere, the solar 27-day cycle signal in daily averaged temperature is more obvious in winter than in summer. In the northern hemisphere, the 27-day signature in temperature at low latitudes (below 50°) for the altitude of 35 – 60 km is more significant in summer than in winter, but for the altitude of 20 – 30 km the signature is more significant in winter. At high latitudes (70 – 85° N), the 27-day signature more significant in winter than in summer, especially for the middle stratosphere (30 – 40 km). In total, the low fraction (less than 10 %) region is larger for "summer" months (October – March) than "winter" months (April – September) for the global region.

An important finding is that large differences exist between northern hemisphere winter and summer. For northern summer (see panel (b) of Figure 11), the latitude-altitude ranges with fractions less than $10\%$ – indicative of a likely solar origin of the identified signatures – are significantly larger than for northern winter (see panel (a) of Figure 11). These differences could be related to enhanced planetary wave activity during northern hemisphere winter, leading to enhanced overall atmospheric variability and consequently making the identification of a solar 27-day signature in atmospheric temperature more difficult.

### 4.1.4 Dependence of the results on solar activity

In addition, we investigated the dependence of the results on solar activity. The comparison of the strong solar activity years (2011 – 2014) with the weak solar activity years (2007 – 2009) is shown in Figure 12 (a – b). The input parameters used here are identical with the ones for Figure 9 (a). The low fraction (less than 10 %) region is larger for strong solar activity years than for weak solar activity years. For weak solar activity years, the low fraction region mainly concentrates in the equatorial mesopause region as shown in Figure 12 (b). For strong solar activity years, the low fraction region is more distributed over high latitudes, mainly at 70 – 85° N and 40 – 60° S at around 40 km, and at 70 – 85° S at around 80 – 80 km.





The results demonstrate that the overall significance of the potential solar 27-day signatures in temperature is generally much lower for solar minimum conditions (see panel (b) of Figure 12) than for solar maximum conditions (see panel (a) of Figure 12). An exception is the tropical mesopause region, where the fraction of random realizations with amplitudes exceeding the amplitude of the actual SEA is smaller for low solar activity than for enhanced solar activity. The reasons for this behaviour

are currently not understood. The general decrease of the significance with decreasing solar activity is, however, as expected. It is also worth pointing out that the overall significance of the results (as quantified by the latitude-altitude ranges with fractions less than 10%) is smaller for enhanced solar activity compared to analyzing the entire data set (compare panel (a) of Figure 12 and panel (a) of Figure 9). This can be explained by the reduced number of epochs available if only parts of the time series are analyzed and highlights the importance of the length of the time series for obtaining statistically significant results.

## 4.2   Sensitivity analysis

The temperature sensitivity to solar forcing was calculated with the method described in section 3.3. Similar to the significance testing, we also investigated the dependence of the sensitivity results on different input and observational parameters.

### 4.2.1   Dependence of the results on statistical parameters

The sensitivity analysis was performed first with the temperature data at the mesopause (88 km) and in the tropics (5° N). Table

2 lists the sensitivity values (i.e., the slope of fitted linear regression line) and the uncertainties depending on the different input parameters. The underlined values in the table represent the maximum and minimum sensitivity values for different cases. The uncertainties are below 0.6 K $(100\ \mathrm{sfu})^{-1}$. The maximum of the sensitivity is 2.74 ($\pm$ 0.28) K $(100\ \mathrm{sfu})^{-1}$ for daily averaged temperature, 3.18 ($\pm$ 0.40) K $(100\ \mathrm{sfu})^{-1}$ for daytime temperature, and 2.95 ($\pm$ 0.45) K $(100\ \mathrm{sfu})^{-1}$ for nighttime temperature. The minimum of the sensitivity is 1.82 ($\pm$ 0.27) K $(100\ \mathrm{sfu})^{-1}$ for daily averaged temperature, 1.33 ($\pm$ 0.34) K $(100\ \mathrm{sfu})^{-1}$ for

daytime temperature, and 1.81 ($\pm$ 0.38) K $(100\ \mathrm{sfu})^{-1}$ for nighttime temperature. The max-to-min variation of the sensitivity value due to different input parameters is 0.92 K $(100\ \mathrm{sfu})^{-1}$ for daily averaged temperature, 1.85 K $(100\ \mathrm{sfu})^{-1}$ for daytime temperature, and 1.14 K $(100\ \mathrm{sfu})^{-1}$ for nighttime temperature. Thus, the influence of the input parameters on the sensitivity result is relatively smaller in daily averaged temperature. This feature is in line with the results derived from the significance test which was discussed in section 4.1.1.

Overall, there is a tendency to larger sensitivity values if a wider window width is used for determining the anomalies. The effect is particularly pronounced for the cases with a 0-day and 7-day smoothing of the anomalies. This dependence of the sensitivities on window width may be expected, because, for narrower window widths, parts of the 27-day signatures present may be removed. The same window width is, however, also used for determining the MgII index anomalies so that part of this effect is compensated, reducing the effect of window width on the sensitivity value. It is also worth pointing out that, for most

cases, the sensitivity values for the different window widths agree within combined uncertainties.



### 4.2.2 Dependence of the results on latitude

Next, we performed the sensitivity analysis for the daily averaged temperature from 2005 to 2017 for latitudes from 85° S to 85° N and altitudes from 20 to 90 km. For this analysis the local solar maxima used in the SEA were determined based on the 7-day smoothed temperature anomalies obtained by subtracting a 35-day running mean from the temperature data. The

resulting sensitivity values and shifts (time lag) are displayed in Figure 13. The obtained sensitivity values range from $-0.02$ to 5.34 K $(100 \text{ sfu})^{-1}$. There are two distinct features in the top panel of Figure 13. First, the sensitivity generally increases with increasing altitude at low latitudes. Second, the higher sensitivity values appear near the poles. Near the equator the sensitivity ranges from $\sim 0$ to 2.80 K $(100 \text{ sfu})^{-1}$, but the maximum sensitivity occurs at 85° N for an altitude of about 40 km. In addition, two distinct features are present in the 70 – 80 km altitude range for southern high latitudes and around 65 km at 40° N.

When comparing the graph with the significance test results shown in Figure 9 (a), it is apparent that the larger sensitivity values appear in regions with lower fraction, i.e., higher significance. The bottom panel of Figure 13 shows the determined time lag between local solar maximum (at the 27-day scale) and the temperature maximum. Comparing the two panels of Figure 13 shows that large time lags tend to occur in latitude-altitude regions with small sensitivity.

In Figure 14 (a) we show the MLS temperature sensitivity to 27-day solar forcing as a function of altitude for a latitude

of 5° S. In order to compare our results to the model calculations based on the three-dimensional chemistry-climate model HAMMONIA analyzed by Gruzdev et al. (2009) (Figure 12 (b) of their paper), we converted the sensitivity to % change in temperature per % change in solar 205 nm irradiance. The conversion is based on a linear fit between the Mg II index and the solar 205 nm irradiance measured by SORCE/SOLSTICE during the period from 2005 to 2017 (LISIRD, 2019), i.e., $\Delta$MgII / $\Delta$205 = 18.928 Mg II index unit $(\text{W m}^{-2} \text{ nm}^{-1})^{-1}$. The percent temperature changes were determined using the mean

temperature of 2005 – 2017 for the latitudes ranging from 85° S to 85° N, and the percent 205 nm irradiance changes were determined using the mean UV 205 nm irradiance between 2005 – 2017. As shown in Figure 14 (a), the maximum is at 84 km and the corresponding sensitivity is 0.13 %/%, a second maximum occurs at 58 km and the corresponding sensitivity is 0.07 %/%. The results are in good agreement with the annually averaged sensitivities for the [20°S, 20°N] latitude range in Gruzdev et al. (2009) (green lines in Figure 12 (b) of their paper). Their model results for enhanced forcing show a main maximum at

85 km and a corresponding sensitivity of about 0.11 %/% and a second maximum at 55 km with a sensitivity of about 0.04 %/%, see black dashed line in Figure 14 (a). For standard forcing, their model results show a main maximum at 85 km and a corresponding sensitivity of about 0.13 %/%, see blue dashed line in Figure 14 (a).

In order to study the sensitivity features for regions with high significance of the identified 27-day signatures, we choose the region that meets the condition that the significance test fraction is less than 10 %. The white parts in the panels of Figure

15 represent the regions for significance test fractions exceeding 10 %. Figure 15 (a) displays the sensitivity and shift of the low fraction region for the latitude range from 85° S to 85° N and the altitude range from 20 to 90 km for years from 2005 to 2017. The red contour lines represent the sensitivity value and the colors represent the shift. The sensitivity is in many cases larger than 1.0 K $(100 \text{ sfu})^{-1}$. The absolute shift is frequently less than 9 days at high altitudes (45 – 90 km). The shift at low altitudes (20 – 45 km) varies largely from $-13$ days to $+13$ days.





### 4.2.3 Dependence of the results on season

Next, the temperature sensitivity to solar forcing was analyzed for different seasons. Figure 11 (c – f) show the sensitivity and shift for the latitude range from 85° S to 85° N and the altitude range from 20 to 90 km for different seasons. As shown in Figure 11 (c – d), the sensitivity in winter is obviously larger than in summer. In the northern hemisphere, the maximum

sensitivity, i.e., 12.41 K $(100 \text{ sfu})^{-1}$, occurs in winter at 85° N for altitudes of about 40 km. In the southern hemisphere, the maximum sensitivity is 5.16 K $(100 \text{ sfu})^{-1}$ and occurs at around 70° S for about 75 km altitude winter. In other words, the sensitivity increases in general with increasing latitude in the winter hemisphere. In summer, the sensitivity shows a tendency to increase with altitude in general. Figure 11 (e – f) shows the determined lag. The shifts do not exhibit the same obvious characteristics as sensitivity, which is not further investigated here.

The graphs indicate larger sensitivity of atmospheric temperature to solar forcing at the 27-day scale in the winter hemisphere (see panels (c) and (d) in Figure 11) – although one has to keep in mind that the results are not significant at all latitudes and altitudes. The identified interhemispheric difference in temperature sensitivity is in agreement with the model results of Gruzdev et al. (2009), who reported that the temperature response to the 27-day solar cycle at extra-tropical latitudes is seasonally dependent with frequently higher sensitivities in winter than in summer. This has also been reported, e.g., by

Ruzmaikin (2007), who analyzed MLS ozone and temperature observations in the stratosphere. The origin of the enhanced sensitivity in the winter hemisphere – particularly at high latitudes – is not well understood.

In Figure 14 (b), we plot the MLS temperature sensitivity profile (%/%) for the southern summer at 75° S (black solid line) and for the northern summer at 75° N (blue line). We used the averaged sensitivity profile (red line) of those two profiles to compare with the results of Thomas et al. (2015) (Figure 8 (b) of their paper), here represented by black dashed line in Figure

14 (b). They analyzed the response of SOFIE temperature observations to the solar 27-day cycle for two northern hemisphere summertime seasons (2010, 2011) and three southern hemisphere (2011-2012, 2012-2013 and 2013-2014) summertime seasons. At 78 km altitude, our sensitivity is 0.13 %/% which is in excellent agreement with the value reported by Thomas et al. (2015). The MLS temperature sensitivity values reported here are larger than the values derived from SOFIE observations for altitudes below 78 km. Our MLS temperature sensitivity is smaller than the SOFIE based values for altitudes above 78 km.

One possible reason for the differences between MLS and SOFIE results could be the different time periods analyzed in the respective studies.

Similar to section 4.2.2, we investigate the sensitivity features for the high significance fraction (< 10 %) region for different seasons as shown in Figure 15 (b – c). The sensitivity is larger than 1.0 K $(100 \text{ sfu})^{-1}$ in most of the high significance region, except for the tropical region at low altitudes (20 – 30 km) for northern winter and southern summer season. In the northern

hemisphere, the large shift of ±13 days appears at around 75 km near the equator in summer, but in winter it occurs at 85° N for an altitude of about 60 km and at 0° – 45° N for the altitude range 20 – 30 km. In the southern hemisphere, a large shift of ± 13 days occurs at low latitudes for the altitude range from 20 – 30 km in summer, but it is mainly focused at high latitudes for the altitude range from 20 to 45 km in winter.





### 4.2.4 Dependence of the results on solar activity

Last, we investigated the dependence of the resulting sensitivity on solar activity. The sensitivity values of the strong solar activity years (2011 – 2014) and the weak solar activity years (2007 – 2009) are shown in Figure 12 (c – d). For strong solar activity years, the sensitivity ranges from $-0.06$ to $7.20$ K $(100$ sfu$)^{-1}$. The sensitivity values are larger at high latitudes than

at low latitudes. In addition, the maximum appears at $85°$ N at about 40 km altitude. The sensitivity values of the strong solar activity years are much smaller than the values in the weak solar activity years. However, unusually high values up to $21.48$ K $(100$ sfu$)^{-1}$ are found for the weak solar activity years, with the maximum occurring at the equatorial mesopause. Such high sensitivities in weak solar activity years likely is an indication that temperature is affected by factors other than the 27-day solar cycle.

Overall, the results show a tendency to enhanced temperature sensitivity to solar forcing during periods of low solar activity. Gruzdev et al. (2009) state that this effect is also present in their model simulations of the effect of the 27-day solar UV forcing on middle atmospheric temperatures, where the sensitivities of temperature to solar activity generally decreases when the forcing increases. For the analysis presented here it is important to remember that for solar minimum conditions the 27-day signatures are not statistically significant at most altitudes and latitudes.

Interestingly, increased sensitivity during period of low solar activity has been reported for 27-day signatures in different atmospheric parameters, including polar summer mesopause temperature (Robert et al., 2010), noctilucent clouds (or polar mesospheric clouds) (Thurairajah et al., 2016) or standard phase heights (von Savigny et al., 2019). These findings may be caused by other sources of variability in a similar period range – likely unrelated to solar forcing – such as planetary wave activity. We refer to von Savigny et al. (2019) for a more detailed discussion on a potential interference by dynamical effects.

Similar to sections 4.2.2 and 4.2.3, the sensitivity and shift for the high significance (i.e., fraction $< 10\,\%$) region for different solar activity are shown in Figure 15 (d – e). For strong solar activity years, a large shift of $\pm 13$ days occurs at southern extra-tropical latitudes for the altitude range from $25 - 45$ km. For weak solar activity years, large shifts of $\pm 13$ days occur at southern extra-tropical latitudes for the altitude range from $80 - 90$ km and at low latitudes for altitudes around 75 km and 20 km.

## 5 Conclusions

This study reports on the investigation of potential solar 27-day signatures in middle atmospheric temperature. The analysis is based on a 13-year (2005 – 2017) global temperature data set obtained from spaceborne measurements with the MLS instrument. The results are mainly based on the superposed epoch analysis approach, which is well suited for identifying weak signatures in time series characterized by large variability. The statistical significance of the obtained results was evaluated with

a dedicated Monte-Carlo approach. The analysis showed that a solar 27-day signature in middle atmospheric temperature can be identified with high statistical significance under certain conditions. However, a complex dependence of the significance of the obtained results on several assumptions and parameters was found. The overall statistical significance of the 27-day signatures is higher for periods of enhanced solar activity than during periods of low solar activity, as expected. The 27-day





signatures in both hemispheres have a higher significance for northern summer compared to northern winter, which may be related to enhanced planetary wave activity during Arctic winters. Several findings indicate the presence of other sources of variability in the 25 – 30 day period range, likely of dynamical nature. The separation of these sources – likely unrelated to solar forcing – from a real solar forcing is an intrinsic difficulty when searching for solar 27-day signatures in atmospheric

parameters. Further studies on the interference of dynamical effects and/or potential solar impact on these dynamical effects are required for a full understanding of the observed variability in middle atmospheric temperature.

*Code availability.*  The source code will be made available by the authors upon request.

*Data availability.*  The data sets used in this paper are publicly accessible. The Bremen daily Mg II index composite data were obtained online from the UV satellite data and science group (http://www.iup.uni-bremen.de/UVSAT/Datasets/mgii, last access: 18 September 2018).

The MLS Level 2 temperature product (version 4.2) was obtained online from the NASA Jet Propulsion Laboratory (https://mls.jpl.nasa.gov/products/temp_product.php, last access: 14 September 2018), The MLS Level 2 Geopotential Height product was obtained online from the NASA data center (https://disc.gsfc.nasa.gov/datasets/ML2GPH_V004/summary?keywords=MLS, last access: 14 September 2018).

*Author contributions.*  PR designed and carried out the tests with the help of CVS. PR prepared the manuscript with contributions from CVS, CZ, CGH, and MJS. CVS provided the code in this study, supervised and guided the analysis process and reviewed the paper. CZ discussed

and reviewed the paper. CGH contributed to the discussion of the method and results and reviewed the paper. MJS is contributor of MLS data, discussed and reviewed the paper.

*Competing interests.*  The authors declare that they have no conflict of interest.

*Acknowledgements.*  This work was supported by the Key Program of National Natural Science Foundation of China (Grant No. 41530422), National High Technology Research and Development Program of China (863 Program) (Grant No.2012AA121101), the Program of Na-

tional Natural Science Foundation of China (Grand No. 61775176) and China Scholarship Council (201706280357). This work was supported by the University of Greifswald. Work at the Jet Propulsion Laboratory, California Institute of Technology, was performed under contract with the National Aeronautics and Space Administration (NASA). We thank the MLS team for providing the high quality MLS data sets, which are the basis of this work. We are indebted to Mark Weber from the University of Bremen for providing the Mg II index data set used in this study.



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

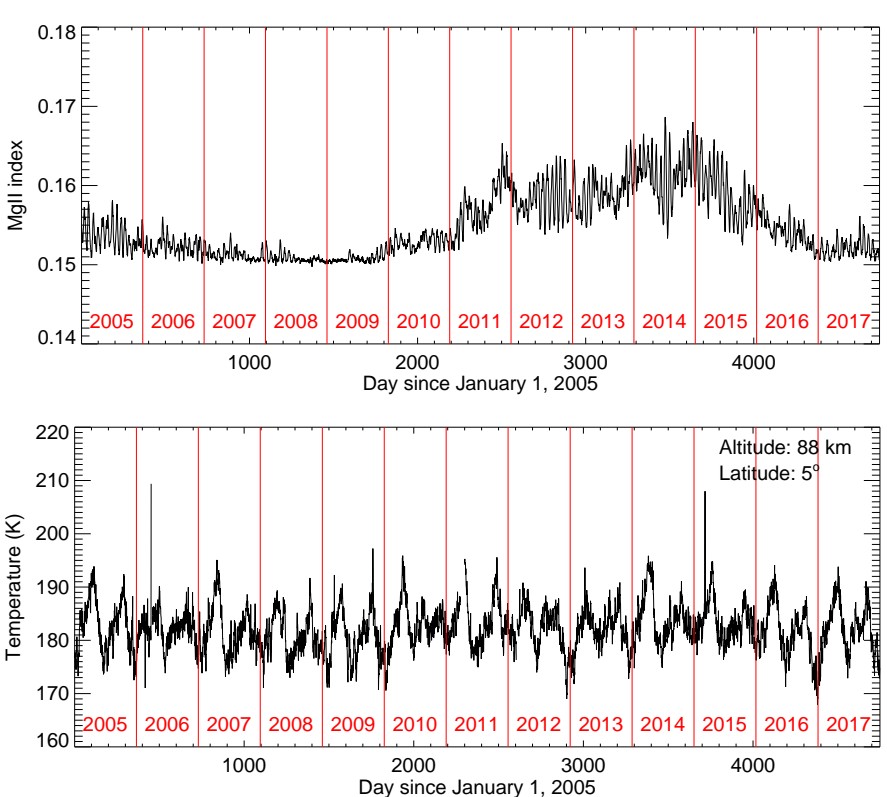

**Figure 1.** Top panel: Mg II index data from 2005 to 2017. Bottom panel: time series of zonally and daily averaged temperature for the 5° N (i.e., 0 – 10° N) latitude bin at 88 km derived from MLS on Aura. Data gaps occur on the days 453–458, 555, 2276–2298, 2605–2609, and 2630–2635.



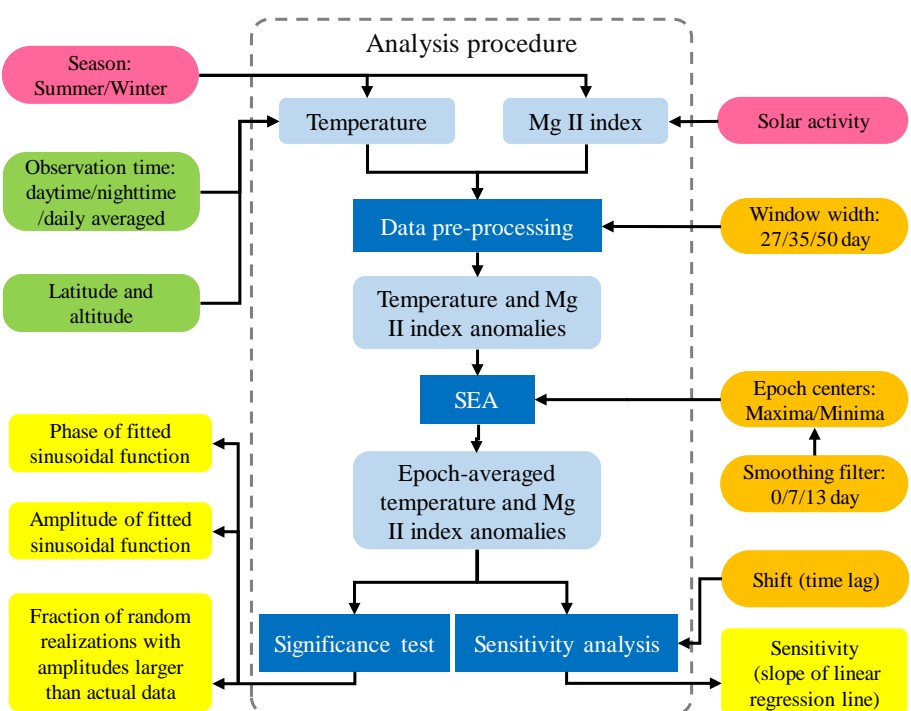

**Figure 2.** Flow chart of the analysis procedure, and the input and output parameters.

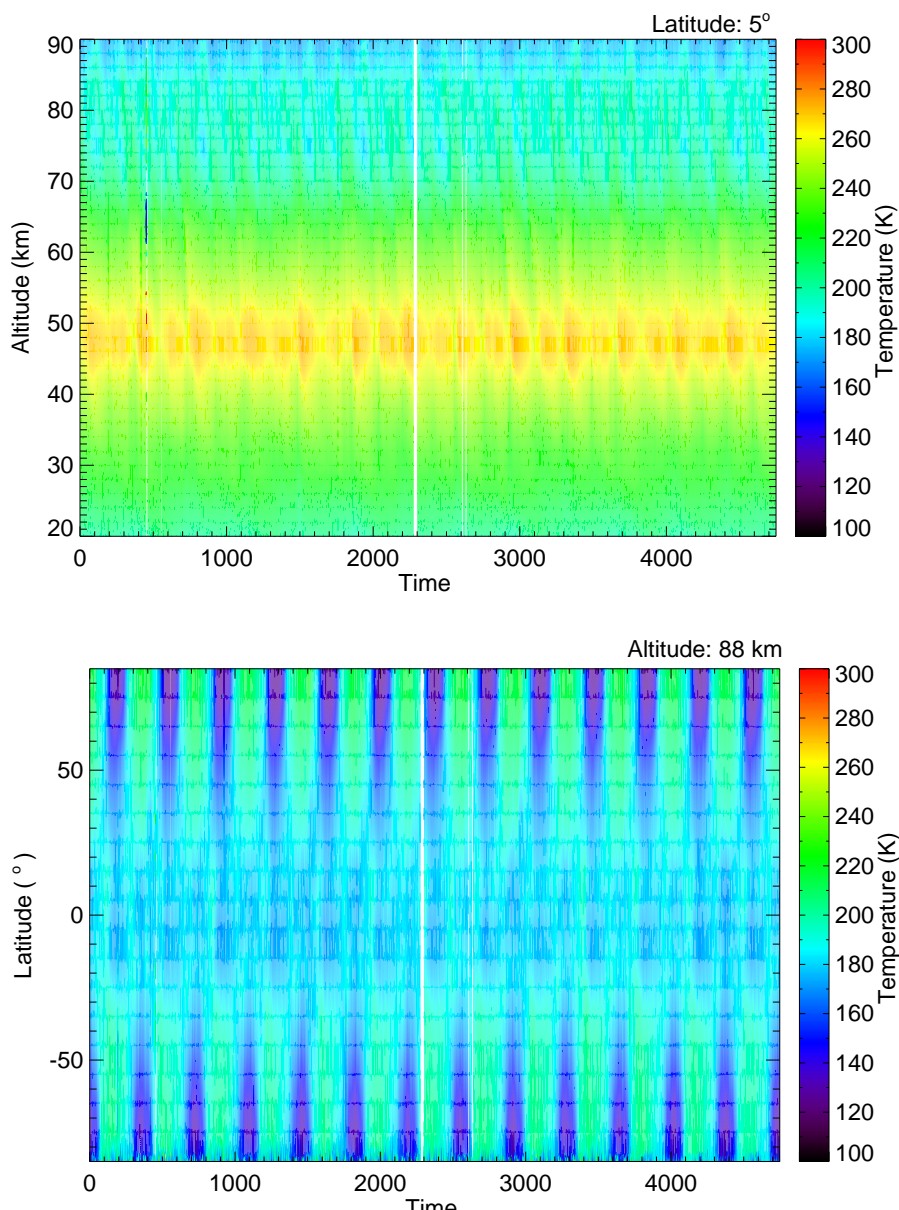

**Figure 3.** Top panel: the MLS temperature as a function of altitude and time day at 5° N. Bottom panel: the MLS temperature as a function of latitude and time at 88 km altitude. The white lines indicate data gaps.

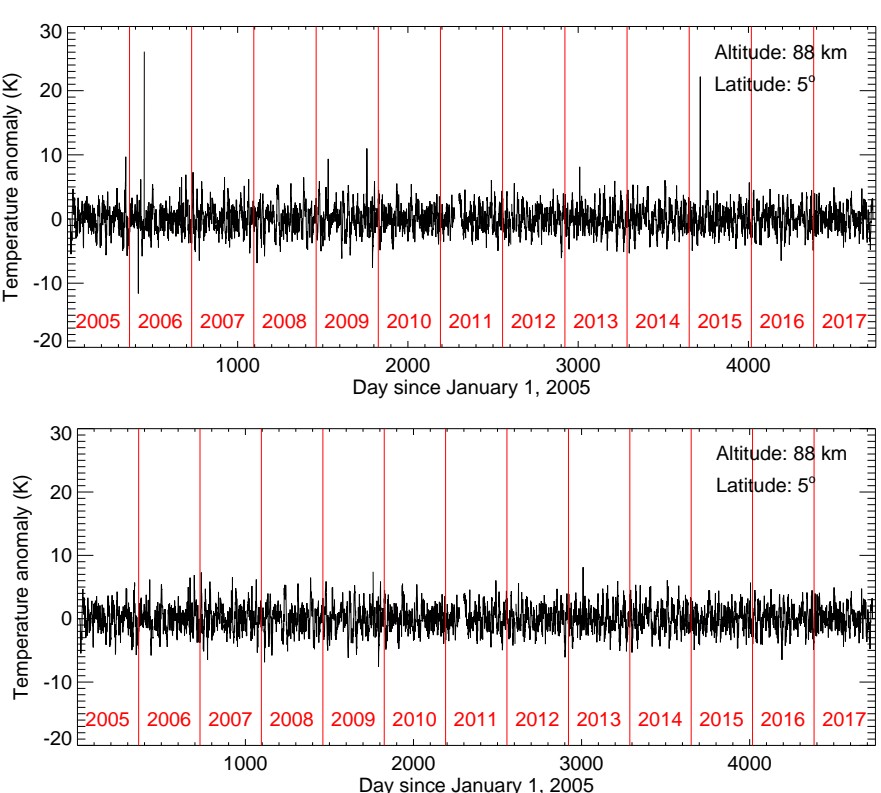

**Figure 4.** Top panel: the MLS temperature anomalies generated by subtracting a 35-day running mean from the time series for an altitude of 88 km and a latitude of 5° N. Bottom panel: Similar to top panel except for avoiding the abnormal peaks on the days 341, 417, 452, 1532, 1759, 3717. The plots are based on daily averaged temperature data.





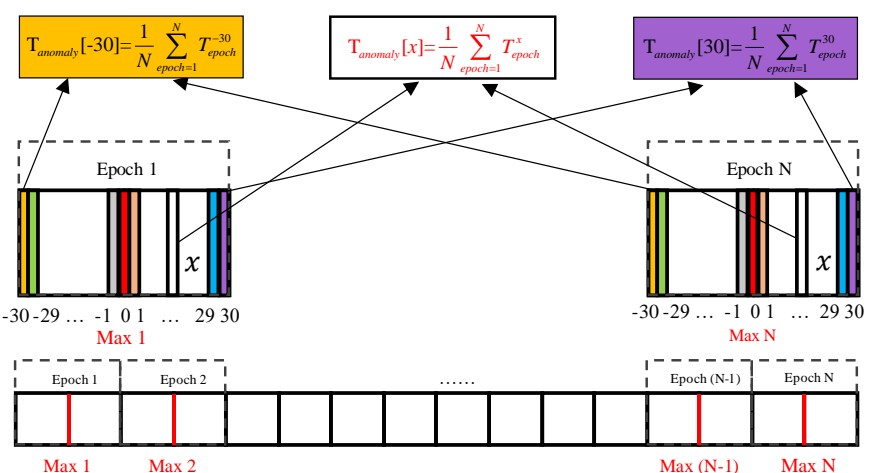

**Figure 5.** Overview of the superposed epoch analysis (SEA). It should be noted that the epochs are allowed to overlap even we do not show it on the figure.



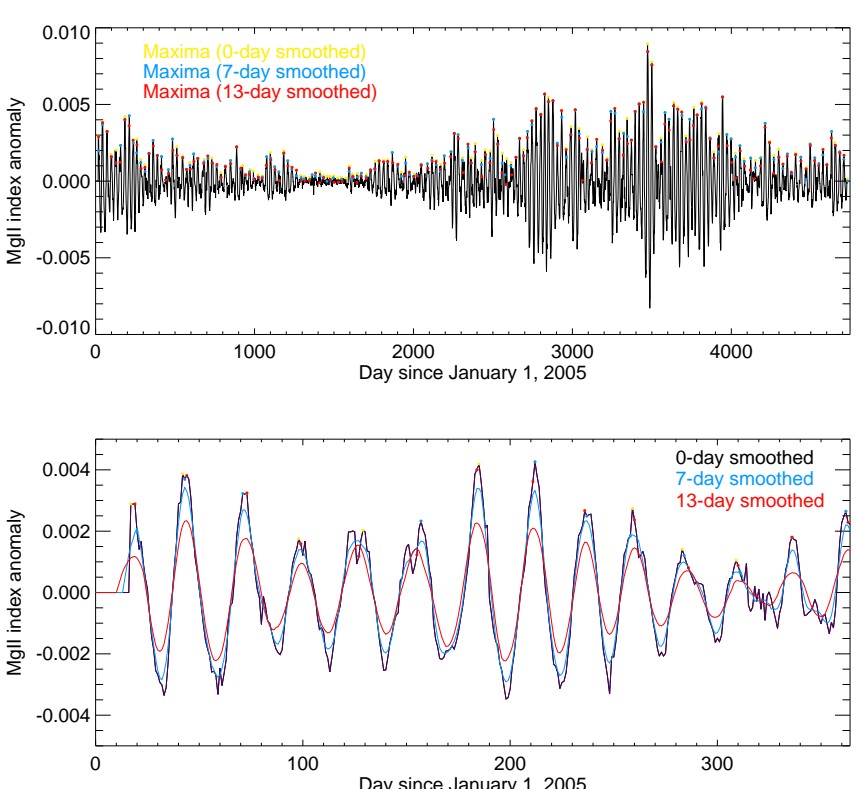

**Figure 6.** Top panel: Mg II index anomalies generated by subtracting a 35-day running mean from the time series. The black line presents the un-smoothed or 0-day smoothed Mg II index anomaly. The yellow, blue and red points are the local maxima chosen from the 0-day, 7-day, and 13-day smoothed Mg II index anomalies, respectively. Bottom panel: similar to top panel except for the year 2005 only. In addition, the blue and red lines present the Mg II index anomalies smoothed by a 7-day and 13-day running mean, respectively.



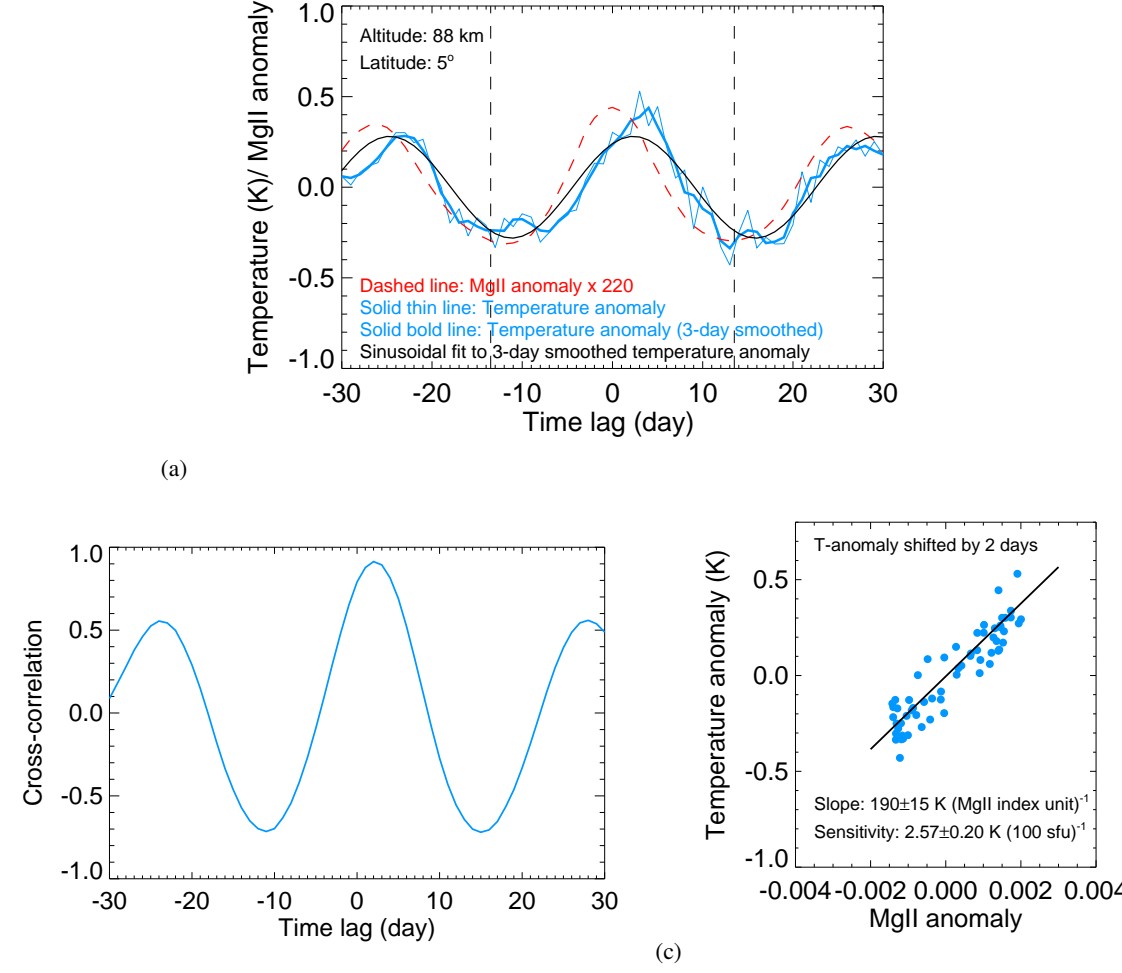

(a)

(b)

(c)

**Figure 7.** (a) Epoch-averaged Mg II index and temperature anomalies for a total of 173 epochs. The dashed red line is the epoch averaged Mg II index anomaly multiplied by a factor of 220. The solid thin blue line corresponds to the epoch averaged temperature anomaly. The solid bold blue line represents the temperature anomaly smoothed with 3-day running mean. The black line is a sinusoidal fit to the 3-day smoothed epoch averaged temperature anomaly, with an amplitude of 0.28 K. (b) Cross correlation between the 61-day epoch-averaged temperature and Mg II index anomaly time series (the results correspond to the 35-day running mean) for the time lag between $-30$ and $+30$ days. (c) Scatter plot of the 2-day lagged temperature and Mg II index anomalies based on the epoch averages displayed in (a). The black line represents the fitted linear regression line.





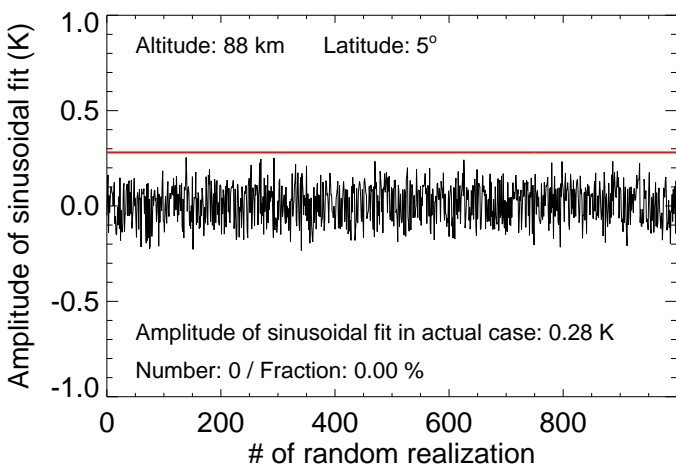

**Figure 8.** Illustration of the Monte-Carlo significance test for an altitude of 88 km and a latitude of 5° N. The red line shows the amplitude of a sinusoidal fit to the extracted 27-day signatures in MLS daily averaged temperature. The black line shows the fitted amplitudes to epoch-averaged temperature anomalies for 1000 randomly chosen epoch ensembles.



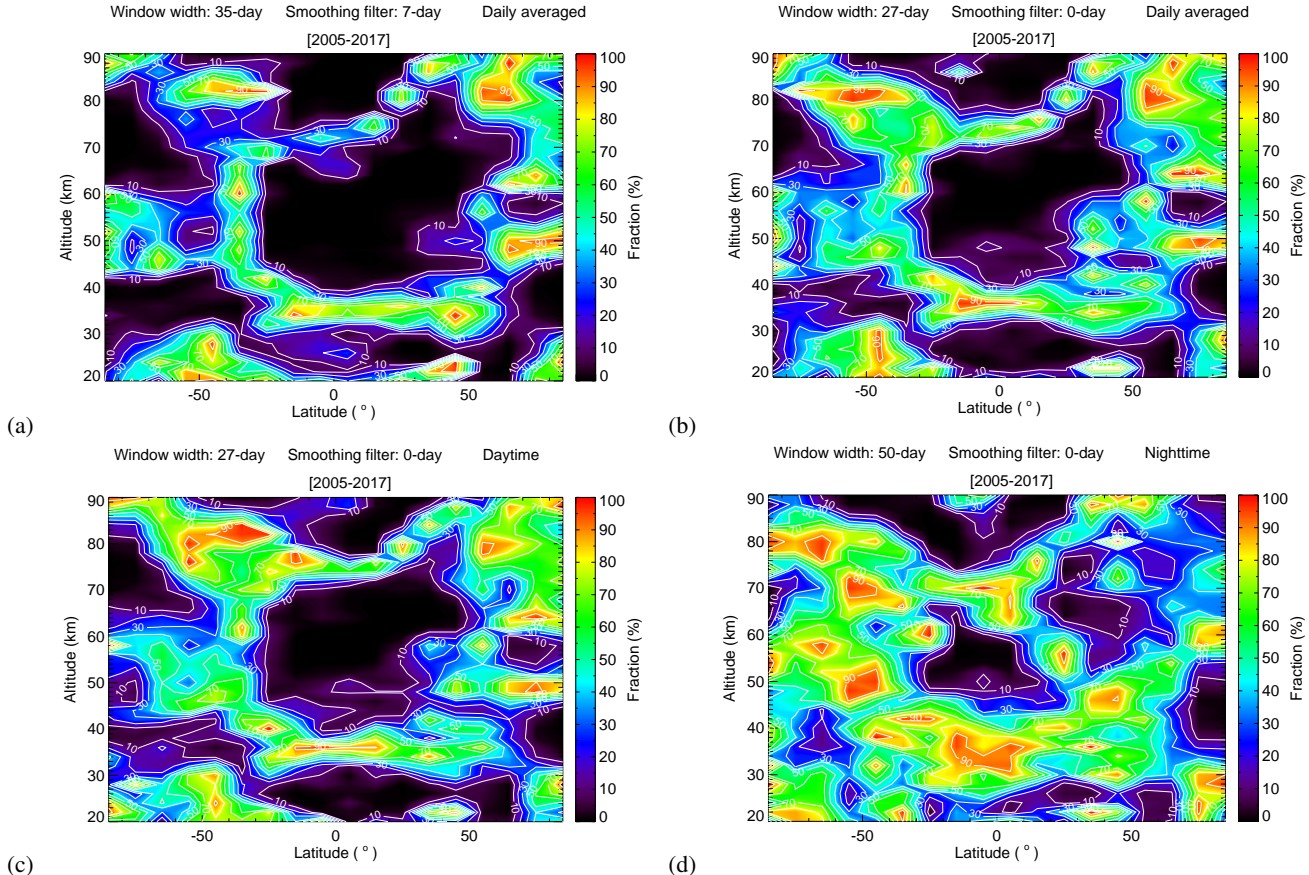

**Figure 9.** (a) The fraction of random realizations with amplitudes larger than the actual SEA based on the daily averaged temperature data for latitudes ranging from 85° S to 85° N and altitudes ranging from 20 to 90 km. A 35-day window width, 7-day smoothing filter and maxima of the Mg II index anomaly are used in this test. (b) Similar to (a) except that a 27-day window width and 0-day smoothing filter are used. (c) Similar to (a) except that a 27-day window width, 0-day smoothing filter and daytime temperature data are used. (d) Similar to (a) except that a 50-day window width, 0-day smoothing filter, nighttime temperature data and minima of the Mg II index anomaly are used.





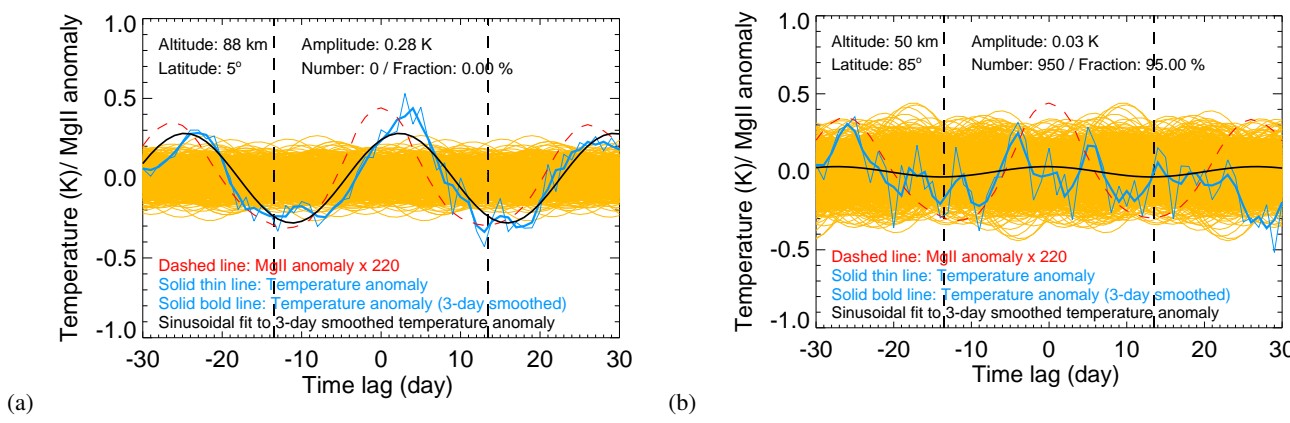

(a)

(b)

**Figure 10.** Similar to Figure 7 (a) except that the orange lines are sinusoidal fit to the 3-day smoothed epoch averaged temperature anomalies for 1000 randomly chosen epoch ensembles.(a) For an altitude of 88 km and a latitude of 5° N (b) For an altitude of 50 km and a latitude of 85° N.





**Figure 11.** (a–b) Similar to Figure 9 (a), except for different seasons. (c–f) Sensitivity and shift for the latitude from 85° S to 85° N and the altitude from 20 to 90 km for different seasons. (a, c, e) are the results for the time range from October to March (northern winter/southern summer), and (b, d, f) are the results for the time range from April to September (northern summer/southern winter).



**Figure 12.** (a–b) Similar to Figure 9 (a), except for strong and weak solar activity years. (c–f) Sensitivity and shift for the latitude from 85° S to 85° N and the altitude from 20 to 90 km for different solar activity.





**Figure 13.** Sensitivity (top) and shift (bottom) for all the latitude from 85° S to 85° N and the altitude from 20 to 90 km, and the analysis year is from 2005 to 2017.

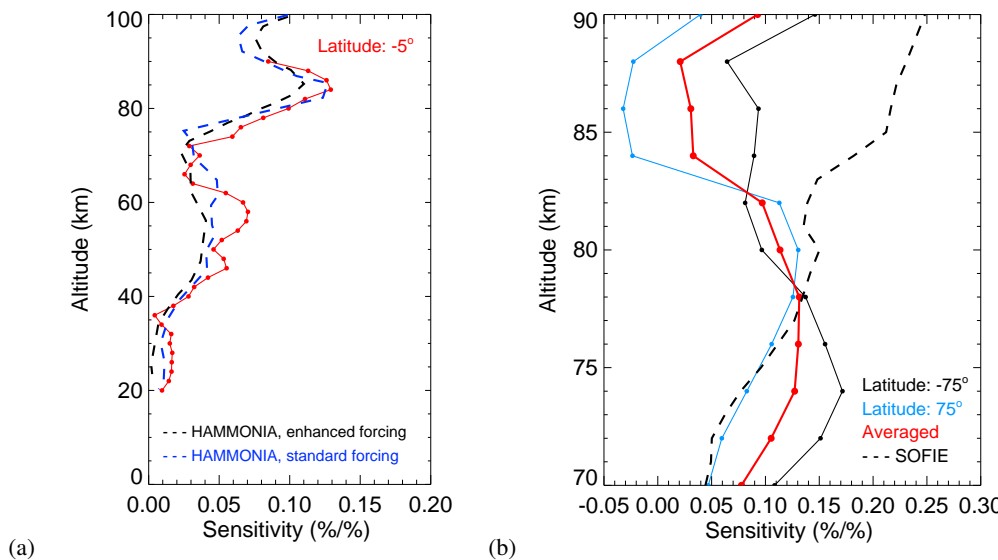

**Figure 14.** (a) MLS temperature sensitivity profile (red line) (sensitivity expressed as % change of temperature per % change in solar UV flux at 205 nm) for a latitude of 5° S and altitudes ranging from 20 to 90 km for the daily averaged temperature data from 2005 to 2017. The profile is from Figure 13 top panel. The dashed lines are the sensitivity results from HAMMONIA for enhanced forcing (black) and for standard forcing (blue) calculated by Gruzdev et al. (2009). (b) Similar to (a), except for the southern summer and a latitude of 75° S (black solid line) and for the northern summer and a latitude of 75° N (blue line) for altitudes from 70 km to 90 km. The sensitivity profile (black solid line) is from Figure 11 (c). The sensitivity profile (blue) is from Figure 11 (d). The red profile is the averaged sensitivities of the black and blue profiles. The black dashed line is the sensitivity results based on SOFIE data from Thomas et al. (2015).





**Figure 15.** Sensitivity (red contour lines) and shift (color filled contour) of the region that satisfies the condition that the significance test fraction less than 10 % for all the latitude from 85° S to 85° N and the altitude from 20 to 90 km for years from 2005 to 2017 (a), for different season (b–c) and for different solar activity (d–e).





**Table 1.** Significance testing results for different input parameters used in the analysis. The temperature data at latitude of 5° N and altitude of 88 km are used here. There are two parameters shown in the table. The first one is absolute amplitude in K of the fitted sinusoidal function. The second one is the fraction (%) of random realizations with amplitudes larger than actual data. The underlined values correspond to the maximum and minimum of the fraction of random realizations with amplitudes larger than the actual data for the daily averaged, the daytime and the nighttime measurements.

| Time series | Temperature | Epoch centers* | Smoothing filter | Window width | | |
|---|---|---|---|---|---|---|
| | | | | 27-day | 35-day | 50-day |
| 2005-2017 | daily averaged | Maxima | 0-day | 0.18 K, 1.0 % | 0.22 K, 0.6 % | 0.22 K, 0.5 % |
| | | | 7-day | 0.21 K, 0.2 % | 0.28 K, 0.0 % | 0.25 K, 0.0 % |
| | | | 13-day | 0.20 K, 0.5 % | 0.23 K, 0.4 % | 0.23 K, 0.3 % |
| | | Minima | 0-day | 0.20 K, 0.5 % | 0.25 K, 0.3 % | 0.22 K, 0.4 % |
| | | | 7-day | 0.20 K, 0.4 % | 0.26 K, 0.2 % | 0.22 K, 0.3 % |
| | | | 13-day | 0.21 K, 0.2 % | 0.26 K, 0.2 % | 0.22 K, 0.3 % |
| | daytime | Maxima | 0-day | 0.14 K, 28.6 % | 0.22 K, 9.8 % | 0.23 K, 4.9 % |
| | | | 7-day | 0.20 K, 6.9 % | 0.27 K, 2.7 % | 0.26 K, 2.4 % |
| | | | 13-day | 0.15 K, 23.4 % | 0.18 K, 20.0 % | 0.16 K, 24.1 % |
| | | Minima | 0-day | 0.23 K, 3.1 % | 0.30 K, 1.5 % | 0.28 K, 1.7 % |
| | | | 7-day | 0.22 K, 3.6 % | 0.27 K, 3.2 % | 0.18 K, 17.4 % |
| | | | 13-day | 0.18 K, 11.4 % | 0.23 K, 8.8 % | 0.17 K, 22.0 % |
| | nighttime | Maxima | 0-day | 0.20 K, 0.8 % | 0.25 K, 0.4 % | 0.23 K, 1.6 % |
| | | | 7-day | 0.24 K, 0.0 % | 0.32 K, 0.0 % | 0.28 K, 0.0 % |
| | | | 13-day | 0.25 K, 0.0 % | 0.31 K, 0.0 % | 0.30 K, 0.0 % |
| | | Minima | 0-day | 0.25 K, 0.1 % | 0.27 K, 0.3 % | 0.22 K, 1.9 % |
| | | | 7-day | 0.23 K, 0.1 % | 0.28 K, 0.1 % | 0.27 K, 0.1 % |
| | | | 13-day | 0.25 K, 0.1 % | 0.30 K, 0.1 % | 0.28 K, 0.1 % |

(*Maxima / Minima of Mg II index anomaly)





**Table 2.** Sensitivity (Unit: $\text{K}\,(100\,\text{sfu})^{-1}$) and the uncertainties of different cases at latitude of 5° N and altitude of 88 km. The sensitivity value is linear fitted by the time lagged epoch averaged temperature anomaly with epoch averaged Mg II index anomaly. The underlined values correspond to the maximum and minimum of the sensitivity value for the daily averaged, the daytime and the nighttime measurements.

| Time series | Temperature | Epoch centers* | Smoothing filter | Window width | | |
|---|---|---|---|---|---|---|
| | | | | 27-day | 35-day | 50-day |
| 2005-2017 | daily averaged | Maxima | 0-day | 1.82 ± 0.27 | 1.91 ± 0.25 | 2.47 ± 0.33 |
| | | | 7-day | 2.44 ± 0.26 | 2.57 ± 0.20 | 2.74 ± 0.28 |
| | | | 13-day | 2.02 ± 0.34 | 1.88 ± 0.26 | 2.25 ± 0.28 |
| | | Minima | 0-day | 2.01 ± 0.27 | 2.29 ± 0.25 | 2.48 ± 0.30 |
| | | | 7-day | 1.91 ± 0.20 | 2.29 ± 0.17 | 2.48 ± 0.24 |
| | | | 13-day | 2.17 ± 0.27 | 2.22 ± 0.23 | 2.08 ± 0.28 |
| | daytime | Maxima | 0-day | 1.91 ± 0.35 | 2.10 ± 0.31 | 2.91 ± 0.40 |
| | | | 7-day | 2.37 ± 0.35 | 2.77 ± 0.30 | 3.18 ± 0.40 |
| | | | 13-day | 1.45 ± 0.48 | 1.33 ± 0.34 | 1.55 ± 0.40 |
| | | Minima | 0-day | 2.27 ± 0.51 | 2.92 ± 0.41 | 2.91 ± 0.51 |
| | | | 7-day | 2.29 ± 0.47 | 2.57 ± 0.36 | 2.10 ± 0.46 |
| | | | 13-day | 2.20 ± 0.38 | 2.21 ± 0.34 | 1.92 ± 0.35 |
| | nighttime | Maxima | 0-day | 1.81 ± 0.38 | 1.96 ± 0.36 | 2.30 ± 0.47 |
| | | | 7-day | 2.55 ± 0.37 | 2.51 ± 0.31 | 2.50 ± 0.44 |
| | | | 13-day | 2.49 ± 0.47 | 2.47 ± 0.36 | 2.89 ± 0.45 |
| | | Minima | 0-day | 1.96 ± 0.48 | 2.46 ± 0.32 | 2.60 ± 0.39 |
| | | | 7-day | 2.06 ± 0.33 | 2.48 ± 0.29 | 2.95 ± 0.45 |
| | | | 13-day | 2.29 ± 0.30 | 2.48 ± 0.25 | 2.57 ± 0.32 |

(*Maxima / Minima of Mg II index anomaly)