# Peer review of "Response of middle atmospheric temperature to the solar 27-day cycle: an analysis of 13 years of MLS data"

_Atmospheric Chemistry and Physics, 2019_

## Referee Comment (RC1) · Anonymous Referee #1 · 9 Oct 2019

General Comments:

Review of 'Response of middle atmospheric temperature to the solar 27-day cycle: an analysis of 13 years of MLS data" by Rong et al., submitted to Atmospheric Chemistry and Physics. The paper is well written and presents interesting results. Minor comments are detailed below.

Specific Comments:

1. Abstract, Line 9: If the sensitivity is larger at high latitudes the why show low latitudes results? Is there a rationale behind choosing 5N for the figures and discussion of results in for e.g. section 4.1.1?

[Figure]

2. Page 2, Line 15-20: Please specify results from Hood et al. (1991) and Brasseur (1993)

3. Page 3; Line 8: Yes, the processes leading to the observed 27-day signatures are not well understood. This manuscript also only speculates about the influence of dynamics but as in other publications doesn't provide any analysis to understand these processes.

4. Page 6; Line 1: Is the 0-day, 7-day, 13-day smooth also applied to the temperature anomalies (page 8; line 1) or is it used only to identify maxima and minima? Figure 7 shows that 3-day smooth is applied to temperature, is the same 3-day data smooth applied to MgII anomaly?

5. Page 12; Line 25: Another reason could be the vertical resolution between MLS (>10 km) and SOFIE (∼2 km). Also, SOFIE measures a range of latitudes (∼65-85).

6. Page 13; Line 13-14: If the 27-day signatures in solar minimum conditions are not statistically significant at most altitudes and latitudes, is the comparison of sensitivity values between strong and weak activity years valid? Can the authors specify what are the altitudes and latitudes that have significant sensitivities?

Technical Corrections:

7. Abstract, line 8: Is there a typo in this sentence "A tendency to higher temperature sensitivity to solar forcing in the winter hemisphere is found" (quantify tendency?)

8. Page3, Lines 3-4: I think there is a typo, grammar issues and/or or missing words in "Besides, an influence of 27-day variability on tropospheric parameters is also debated". Maybe the authors are saying that the 27-day variability on tropospheric parameters has been studied previously?

9. Page 3; line 27: typo "...radio flux 'or' can be...

10. Page 9, line 17: typo – "...the 27-day signature 'is' more significant..."

---

## Referee Comment (RC2) · Mark Weber (Referee) · 23 Oct 2019

**1   General comment**

This paper reports on the 27-day (corresponding to one solar rotation) periodicities in temperature data from MLS covering 13 years of data and the vertical range 20-90 km. The authors use the super-epoch approach to identify the 27 day signal. Various sensitivity tests are made to demonstrate the robustness of their results. The scientific methods they use is is very clearly described and their work tracable for the reader. I recommend publication after some minor corrections. My main criticism is

that an evaluation of the results obtained here with respect to other similar analyses on temperature data (many cited and summarised in the Introduction) remains somewhat vague. It is important to stress here which results are new (not seen by others) in addition to confirmation of agreement with prior work.

**2  Detailed points**

Page 2, l. 3: "While a significant number of experimental studies investigated solar-driven 27-day variations in stratospheric and mesospheric parameters, the physical/chemical mechanisms leading to these signatures are, in many cases, not well understood. Therefore, it has become a highly interesting subject to study atmospheric variations due to the 27-day solar activity cycle in middle atmospheric parameters." The expectation is raised here that additional studies (like this) will lead to a better understanding of the processes behind the 27d variability. However, this study is simply another study lining up with others on fingerprint detection, but falls short of identifying the processes behind these changes (except for some plausibility arguments that the processes may be dynamical rather than direct solar in nature)

p. 13: Conclusions: here it may be important to briefly summarise what are the new findings from this study with respect to earlier work (see general comment)

p. 3, l. 25: Here one should briefly mention why Mg II (and not F10.7 or Ly-alpha) is used here. Dudok de Wit et al. (2009) and others have shown that the Mg II best correlates with solar UV radiation variation particularly during solar minimum conditions. The translation of Mg II changes into equivalent F10.7cm flux and 205 nm irradiance is needed since other studies used the latter.

p. 3 l. 32: "derived from four data sets". Other satellite data were used to fill the gap.

The fifth major dataset is the early SBUV record (before 1995, not relevant here).

p. 4, l. 11: "MLS version 4.2 temperature is" –> "MLS temperatures are". Version 2.4 is already mentioned in the sentence before.

p. 5., l. 12: Figure 3 is mentioned before Figure 2 in the main text. Please check.

p. 6., l. 5: "Second" –> "Secondly"

Figure 8, l. 29: Wouldn't it be better to invert the color scale in the significance plots (Figure 9 and other figures). That way those regions are highlighted (and more colorful) where the significance of the 27d signal is high! For the axis label I would use "(statistical) significance" rather than "fraction". It would be also useful to shade out regions where no statistical significance is given. This helps to focus on the relevant part in the plots. This applies to Figs. 9, 11, and 12.

Section 4.1.3 (p. 9) Discussion on the time lag plots (Figure 11) is missing.

p. 9, l. 32: Suggest to use "region of high significance" rather than "low fraction region" throughout the main text.

p. 10, l. 16: "different input parameters". Better say "different settings", as input data (MLS, Mg II) remain the same.

p. 11, l. 10: "When comparing the graph with the significance test results shown in Figure 9 (a), it is apparent that the larger sensitivity values appear in regions with lower fraction, i.e., higher significance." If obvious, why mention it here (can be omitted).

p. 11, l. 11: "determined time lag" –> "time lag"

p. 11, l- l2: "Comparing the two panels of Figure 13 shows that large time lags tend to occur in latitude-altitude regions with small sensitivity". Why mention ii here (if greyed out in the plot because of no significance)

p. 12, l. 9: "obvious characteristics". What are they?

p. 12, l. 27: "high significance fraction" –> "high significance region"

Figure 6: Yellow curve is hard to see, suggest to plot first the unsmoothed curve and then overplot this with the smoother curves (order: red, blue, yellow). Then all curves may be visible. Yellow is hard to see, use another color for better legibility.

Figure 7: Show a vertical line in panel b, indicating the derived time lag.

---

## Author Comment (AC1) · 7 Dec 2019

General Comments:
Review of 'Response of middle atmospheric temperature to the solar 27-day cycle: an analysis of 13 years of MLS data" by Rong et al., submitted to Atmospheric Chemistry and Physics. The paper is well written and presents interesting results. Minor comments are detailed below.

> ***Answer:***
>
> ***We thank the reviewer for this encouraging comment.***

Specific Comments:
1. Abstract, Line 9: If the sensitivity is larger at high latitudes the why show low latitudes results? Is there a rationale behind choosing 5N for the figures and discussion of results in for e.g. section 4.1.1?

> ***Answer:***
>
> ***In general, from Figure 13 (top panel), we could draw the conclusion that the sensitivity is larger at high latitudes. We chose the zonally averaged temperature at 88 km for the [0-10° N] latitude range as subject to discuss for the following two reasons:***
>
> ***One reason is that von Savigny et al. (2012) reported the sensitivity of equatorial mesopause temperature to the 27-day solar cycle. They analyzed zonally averaged OH(3-1) rotational temperatures at 87 km for the [0, 20° N] latitude range using the Mg II index derived from SCIAMACHY. Choosing the close location to analyze is good to compare our results with theirs.***
>
> ***Another reason is that Figure 13 (top panel) shows that the high sensitivity also occurs at the mesopause of low latitudes. Therefore, it appears reasonable to choose 5 ̊N at 88 km for the discussion.***

2. Page 2, Line 15-20: Please specify results from Hood et al. (1991) and Brasseur (1993)

> ***Answer:***
>
> ***OK, we specified results from Hood et al. (1991) and Brasseur (1993) in the manuscript.***
>
> > ***Changes to paper:***
> >
> > ***(page 2, line 14-30) "***

*(Stratosphere and Mesosphere Sounder) measurements at low latitudes to determine the temperature sensitivity to solar forcing at the 27-day scale for altitudes extending up to about 90 km.* **Hood (1986)** *used Nimbus-7/SAMS temperature measurements (December 24, 1978 to May 20, 1981) at low latitudes (25° S to 25° N) to determine the temperature sensitivity to solar forcing at the 27-day scale for altitudes ranging from about 24 km to 57 km, yielding a maximum temperature response amplitude of 0.36 % (~1 K) near the stratopause. The peak-to-peak variations in the 205-nm flux were as large as 6% on the 27-day time scale during their study period. Later,* **Hood et al. (1991)** *presented an analysis of 4.3 years (December 24, 1978 to June 9, 1983) of Nimbus-7/SAMS temperature data for estimating and characterizing the response of mesospheric temperature to solar ultraviolet variations at the 27-day scale. They found that the maximum low-latitude temperature response amplitudes (approximately 1.3 K for the maximum observed Lyman-$\alpha$ flux change of ~29%) occur at a level of ~ 0.06 mbar, approximately 68 km altitude, in agreement with Keating et al. (1987).* **Brasseur (1993) used a two-dimensional chemical-dynamical-radiative model of the middle atmosphere to investigate the potential changes of temperature in response to the 27-day variation in the solar ultraviolet flux.** *They found that the largest temperature response amplitude (approximately 0.37 K) is at the stratopause corresponding to a peak-to-trough solar variation of 3.3% at 205 nm. The temperature sensitivity using their model for equatorial regions is 0.01 K/% at 30 km, 0.06 K/% at 40 km, and 0.12 K/% at 60 km, and the modelled sensitivity for altitudes ranging from 40 km to 60 km is in agreement with Keating et al. (1987).* **The temperature response to solar variability has not been considered at altitudes above 60 km in Brasseur (1993), because several radiative processes specific to the mesosphere had not been treated in detail."**

3. Page 3; Line 8: Yes, the processes leading to the observed 27-day signatures are not well understood. This manuscript also only speculates about the influence of dynamics but as in other publications doesn't provide any analysis to understand these processes.

*Answer:*

*The point is similar to the first point of detailed points raised by reviewer #2. We changed the sentence to avoid the misleading to readers. The reviewer is certainly absolutely correct that our manuscript does not really attempt to explain the underlying physico-chemical processes. This is beyond the scope of our study and will require dedicated model simulations taking all relevant physical and chemical processes into account.*

*Changes to paper:*

*(page 3, Line 16-18) "~~In brief, previous studies found that variations of solar spectral irradiance at the 27-day time scale affect atmospheric temperature based on different observational and modelling data sets. However, for many atmospheric species and parameters, the processes leading to the observed 27-day signatures are not well understood. In addition, the statistical significance of the identified signatures is often difficult to establish.~~ While the works cited above have found correlations between 27-day variations of solar spectral irradiance and atmospheric temperature variability in numerous observational and modelling data sets, there is still work to be done in characterizing and quantifying the significance of observed 27-day signatures."*

4. Page 6; Line 1: Is the 0-day, 7-day, 13-day smooth also applied to the temperature anomalies (page 8; line 1) or is it used only to identify maxima and minima? Figure 7 shows that 3-day smooth is applied to temperature, is the same 3-day data smooth applied to MgII anomaly?

*Answer:*

*Question (1):* Page 6; Line 1: Is the 0-day, 7-day, 13-day smooth also applied to the temperature anomalies (page 8; line 1) or is it used only to identify maxima and minima?

*We apologize that the manuscript is indeed a bit incorrect regarding this point. The 0-day, 7-day, 13-day smoothing is actually only applied to the Mg II index anomalies (not epoch-averaged) to identify maxima and minima, and is not applied to the temperature anomalies. We changed three places in the manuscript as follows.*

*Changes to paper:*

*(page 8, line 16-18) "as well as different local maxima chosen by 0-day/7-day/13-day smoothed  Mg II index anomalies , for daytime, nighttime, and daily averaged temperature observations"*

*(page 9, line 4-7) "For the results shown in Figure 9 (a), the local solar maxima used in the SEA are chosen from the 7-day smoothed  Mg II index anomalies obtained by subtracting a 35-day running mean from the  Mg II index data. The temperature anomalies used in the SEA are obtained by subtracting a 35-day running mean from the daily averaged temperature time series."*

*(page 11, line 21-24) "For this analysis the local solar maxima used in the SEA were determined based on the 7-day smoothed  Mg II index anomalies obtained by subtracting a 35-day running mean from the  Mg II index data. The temperature anomalies used in the SEA are obtained by subtracting a 35-day running mean from the daily averaged temperature time series."*

*Question (2):* Figure 7 shows that 3-day smooth is applied to temperature, is the same 3-day data smooth applied to MgII anomaly?

*In Figure 7, the 3-day smoothing is only applied to epoch-averaged temperature anomalies to calculate the sinusoidal fit. The obtained amplitude of the sinusoidal fit of the 3-day smoothed epoch-averaged temperature anomalies is used to perform the significance test calculations. For the significance test, the 3-day smoothing is applied both to the 1000 random cases and to the actual case.*

*For the sensitivity analysis, the same 3-day smoothing is not applied to epoch-averaged temperature anomalies or epoch-averaged Mg II index anomalies. The sensitivity parameter (the slope k) shown in Figure 7 (c) is based on the linear regression line to the data points, i.e., un-smoothed epoch-averaged temperature and Mg II index anomalies.*

*To clarify, we changed three places in the manuscript as follows.*

*Changes to paper:*

*(page 6, line 27-29) "The obtained epoch-averaged temperature and Mg II index anomalies (un-smoothed) are used in the sensitivity analysis. A 3-day smoothing is applied to epoch-averaged temperature and Mg II index anomalies for the significance test."*

*(page 7, line 6) "The sensitivity is directly determined by the slope (k) of a linear regression line to the data points, i.e., un-smoothed epoch-averaged temperature and Mg II index anomalies.*

*(page 7, line 30) "Then a sinusoidal function is used to fit every single random realization of the 3-day smoothed epoch averaged temperature anomaly."*

5. Page 12; Line 25: Another reason could be the vertical resolution between MLS (>10 km) and SOFIE (~2 km). Also, SOFIE measures a range of latitudes (~65-85).

*Answer:*

*Yes, we agree with the reviewer. It can be another possible reason. We added it in the paper.*

*Changes to paper:*

*(page 13, line 13-16) "Another reason could be the difference in vertical resolution between MLS (>10 km) and SOFIE (~ 2 km) for the range of altitudes relevant here (70 - 90 km). Also, the spatial and temporal sampling of the MLS and SOFIE measurements differs, as the latitudes of SOFIE solar occultation measurements vary slowly from day to day within the ~ 65°- 85° NS latitude range."*

6. Page 13; Line 13-14: If the 27-day signatures in solar minimum conditions are not statistically significant at most altitudes and latitudes, is the comparison of sensitivity values between strong and weak activity years valid? Can the authors specify what are the altitudes and latitudes that have significant sensitivities?

*Answer:*

*Question (1):* If the 27-day signatures in solar minimum conditions are not statistically significant at most altitudes and latitudes, is the comparison of sensitivity values between strong and weak activity years valid?

*The reviewer's suggestion is fully correct. The low statistical significance of the results for periods of low solar activity is the reason, why comparisons should be interpreted carefully. This was the intended meaning of the sentence, but this was obviously not very clear. We added the following statement to this paragraph.*

*Changes to paper:*

*(page 14, line 3-4) "For this reason the comparison of sensitivity values for periods of high and low solar activity should be interpreted with caution."*

*Question (2):* Can the authors specify what are the altitudes and latitudes that have significant sensitivities?

*Yes, we specified the region that have significant sensitivities in the text.*

*Changes to paper:*

*(page 14, line 11-14) "The colored areas are the latitudes and altitudes that have significant sensitivities. For solar maximum, the region of high latitude of 85° N at about 40 km has highly significant sensitivities of about 5.0 - 7.2 K (100 sfu)$^{-1}$. For solar minimum, the high altitudes of 80 - 90 km near the equator have highly significant sensitivities of about 17.0 - 21.5 K (100 sfu)$^{-1}$."*

Technical Corrections:

7. Abstract, line 8: Is there a typo in this sentence "A tendency to higher temperature sensitivity to solar forcing in the winter hemisphere is found" (quantify tendency?)

*Answer:*

*This statement is incomplete and we changed it as follows. We hope this is now easier to follow.*

> *Changes to paper:*
>
> *(Abstract, line 8-10) "A tendency to higher temperature sensitivity to solar forcing in the winter hemisphere compared to the summer hemisphere is found."*

8. Page3, Lines 3-4: I think there is a typo, grammar issues and/or or missing words in "Besides, an influence of 27-day variability on tropospheric parameters is also debated". Maybe the authors are saying that the 27-day variability on tropospheric parameters has been studied previously?

> *Answer:*
>
> *Yes, the grammar was changed.*
>
> > *Changes to paper:*
> >
> > *(page 3, line 13-15) " The influence of 27-day variability on tropospheric parameters  has also previously been  discussed (e.g., Hoffmann and von Savigny (2019) and references therein), but this work focuses specifically on the middle atmosphere, so the troposphere is not discussed here. "*

9. Page 3; line 27: typo ": : :radio flux 'or' can be: : :.

> *Answer:*
>
> *Yes, the sentence has be corrected by deleting the extra "or".*
>
> > *Changes to paper:*
> >
> > *(page 4, line 6) "the F10.7 cm radio flux  can be easily established by a linear regression (e.g., von Savigny et al., 2012, 2019)."*

10. Page 9, line 17: typo – ": : :the 27-day signature 'is' more significant: : :"

> *Answer:*
>
> *Yes, this sentence was corrected by adding the missing 'is'.*
>
> > *Changes to paper:*
> >
> > *(page 10, line 5) "At high latitudes ($70 - 85 °N$), the 27-day signature is more significant in winter than in summer, especially for the middle stratosphere ($30 - 40$ km). "*

**REFERENCE**

*Brasseur, G.: The response of the middle atmosphere to longterm and short-term solar variability: A two dimensional model, J. Geophys. Res., 98, 23079 – 23090, https://doi.org/doi:10.1029/93JD02406, 1993.*

*Hood, L. L.: Coupled stratospheric ozone and temperature responses to short-term changes in solar ultraviolet flux: an analysis of Nimbus 7 SBUV and SAMS data, J. Geophys. Res. 91 (D4), 5264 – 5276, https://doi.org/10.1029/JD091iD04p05264, 1986.*

*Hood, L. L., Huang, Z., and Bougher, S. W.: Mesospheric effects of solar ultraviolet variations: Further analysis of SME IR ozone and Nimbus 7 SAMS temperature data, J. Geophys. Res., 96, 12989 – 13002, https://doi.org/10.1029/91JD01177, 1991.*

*Keating, G., Pitts, M., Brasseur, G., and Rudder, A. D.: Response of middle atmosphere to short-term solar ultraviolet variations: 1. Observations, J. Geophys. Res., 92, 889 – 902,*

*https://doi.org/10.1029/JD092iD01p00889, 1987.*

*von Savigny, C., Eichmann, K. U., Robert, C. E., Burrows, J. P., and Weber, M.: Sensitivity of equatorial mesopause temperatures to the 27-day solar cycle, Geophys. Res. Lett., 39, L21804, https://doi.org/10.1029/2012GL053563, 2012.*
This paper reports on the 27-day (corresponding to one solar rotation) periodicities in temperature data from MLS covering 13 years of data and the vertical range 20-90 km. The authors use the super-epoch approach to identify the 27 day signal. Various sensitivity tests are made to demonstrate the robustness of their results. The scientific methods they use is is very clearly described and their work tracable for the reader. I recommend publication after some minor corrections. My main criticism is that an evaluation of the results obtained here with respect to other similar analyses on temperature data (many cited and summarised in the Introduction) remains somewhat vague. It is important to stress here which results are new (not seen by others) in addition to confirmation of agreement with prior work.

> *Answer:*
>
> *We thank the reviewer for his encouraging comments and helpful suggestions. We added several sentences in the conclusion to summarise our new results, the details please see our answer to the second comment of the following detailed points.*

2 Detailed points

Page 2, l. 3: "While a significant number of experimental studies investigated solar-driven 27-day variations in stratospheric and mesospheric parameters, the physical/chemical mechanisms leading to these signatures are, in many cases, not well understood. Therefore, it has become a highly interesting subject to study atmospheric variations due to the 27-day solar activity cycle in middle atmospheric parameters." The expectation is raised here that additional studies (like this) will lead to a better understanding of the processes behind the 27d variability. However, this study is simply another study lining up with others on fingerprint detection, but falls short of identifying the processes behind these changes (except for some plausibility arguments that the processes may be dynamical rather than direct solar in nature)

> *Answer:*
>
> *Yes, this comment is similar to comment 3 of reviewer #1. We apologize that these sentences are misleading and imply another content of this paper. To avoid this problem, we rewrote this part.*
>
> > *Changes to paper:*
> >
> > *(page 2, line 3-5) "While a significant number of experimental studies investigated solar-driven 27-day variations in stratospheric and mesospheric parameters, the  further characteristics of these signatures are yet to be discovered. ."*

p. 13: Conclusions: here it may be important to briefly summarise what are the new findings from this study with respect to earlier work (see general comment)

*Answer:*

*We added several sentences in the conclusion to summarise our new results as follows.*

*Changes to paper:*

*(page 14, Conclusion) "This study reports on the investigation of potential solar 27-day signatures in middle atmospheric temperature. The analysis is based on a 13-year (2005 – 2017) global temperature data set obtained from spaceborne measurements with the Aura MLS instrument. The results are mainly based on the superposed epoch analysis approach, which is well suited for identifying weak signatures in time series characterized by large variability. The statistical significance of the obtained results was evaluated with a dedicated Monte-Carlo approach. On this basis, several new conclusions can be drawn.*

*(1) The analysis showed that a solar 27-day signature in middle atmospheric temperature can be identified with high statistical significance under certain conditions. However, a complex dependence of the significance of the obtained results on several assumptions and parameters was found.*

*(2) The sensitivity of temperature to solar 27-day forcing tends to be larger at high latitudes than at low latitudes.*

*(3) The overall statistical significance of the 27-day signatures is higher for periods of enhanced solar activity than during periods of low solar activity, as expected. The sensitivity analysis showed that even for strong solar activity, the 27-day signatures are not significant at many latitudes and altitudes.*

*(4) Enhanced 27-day signatures during winter were found. It is noteworthy that the 27-day signatures in both hemispheres have a higher significance for northern summer compared to northern winter, which may be related to enhanced planetary wave activity during Arctic winters.*

*Several findings indicate the presence of other sources of variability in the 25 – 30 day period range, likely of dynamical nature. The separation of these sources – likely unrelated to solar forcing – from a real solar forcing is an intrinsic difficulty when searching for solar 27-day signatures in atmospheric parameters. Further studies on the interference of dynamical effects and/or potential solar impact on these dynamical effects are required for a full understanding of the observed variability in middle atmospheric temperature."*

p. 3, l. 25: Here one should briefly mention why Mg II (and not F10.7 or Ly-alpha) is used here. Dudok de Wit et al. (2009) and others have shown that the Mg II best correlates with solar UV radiation variation particularly during solar minimum conditions. The translation of Mg II changes into equivalent F10.7cm flux and 205 nm irradiance is needed since other studies used the latter.

*Answer:*

*Thanks for this information. We added it to the text.*

*Changes to paper:*

*(page 4, line 1-8) "In contrast to other solar proxies (such as the Lyman-α and the F10.7 cm radio flux), the Mg II index is used here because the Mg II best correlates with solar UV radiation variation, particularly during solar minimum conditions (Dudok de Wit et al., 2009; Snow et al., 2014).*

*The Mg II index is a dimensionless proxy. The relationship between the Mg II index and other solar proxies, e.g., the Lyman-$\alpha$ or the F10.7 cm radio flux  can be easily established by a linear regression (e.g., von Savigny et al., 2012, 2019). The F10.7 cm radio flux is usually given in solar flux units (sfu), which are equal to $10^{-22}$ W m$^{-2}$ Hz$^{-1}$.  This allows the results to be compared  with other research results."*

p. 3 l. 32: "derived from four data sets". Other satellite data were used to fill the gap. The fifth major dataset is the early SBUV record (before 1995, not relevant here).

> *Answer:*
>
> *Ok, we changed this sentence as follows.*
>
> > ***Changes to paper:***
> >
> > *(page 4, line 9-12) "For this study we employ the Bremen daily Mg II index composite data as the solar proxy, which is available from 1978 to present and is derived from  six data sets, i.e., the Solar Backscatter UltraViolet (SBUV) (before 1995), the Global Ozone Monitoring Experiment (GOME) (1995-2011), SCIAMACHY (2002-2012), GOME-2A (since 2007),  GOME-2B (since 2012), and GOME-2C (since 2019)."*

p. 4, l. 11: "MLS version 4.2 temperature is" –> "MLS temperatures are". Version 2.4 is already mentioned in the sentence before.

> *Answer:*
>
> *Okay, deleted.*
>
> > ***Changes to paper:***
> >
> > *(page 4, line 24) "In this work, we use the MLS Level 2 temperature product version 4.2. MLS  temperature is available from 2 August 2004 to present."*

p. 5., l. 12: Figure 3 is mentioned before Figure 2 in the main text. Please check.

> *Answer:*
>
> *There is no problem to the number of Figures.*
>
> > *We checked and found the Figure 2 is first mentioned in page 5 line 4 (discussion version: page 4 line 24) and the Figure 3 is first mentioned in page 5 line 24 (discussion version: page 5 line 13).*

p. 6., l. 5: "Second" –> "Secondly"

> *Answer:*
>
> *We would like to keep using "second" (page 6 line 18), because "second" to be a parallel construction with "first" in the previous paragraph (page 6 line 10).*

Figure 8, l. 29: Wouldn't it be better to invert the color scale in the significance plots (Figure 9 and other figures). That way those regions are highlighted (and more colorful) where the significance of the 27d signal is high! For the axis label I would use "(statistical) significance" rather than "fraction". It would be also useful to shade out regions where no statistical significance is given. This helps to focus on the relevant part in the plots. This applies to Figs. 9, 11, and 12.

> *Answer:*
>
> *Yes, the first suggestion is very good and we inverted the color scale in Figs. 9, 11 and 12.*

*However, we would like to retain the axis label as "fraction", because the value of color bar shown here means fraction. We think, using "fraction" would be more straightforward to show what the values are. Besides, statistical significance is opposite to fraction, i.e., the smaller the fraction is, the higher the significance is. We think using "(statistical) significance" may cause a bit of misleading if the value is fraction.*

*We agree with the reviewer that shading out regions where no statistical significance helps to focus on the relevant part in the plots. We have already done a similar thing in Figure 15, only the relevant part (high significant region) was shown in color. We think Figure 15 is already clearly shown the characteristics of the relevant part and is easier to read for readers.*

***Changes to paper:***

[Figure]

[Figure]

Section 4.1.3 (p. 9) Discussion on the time lag plots (Figure 11) is missing.

*Answer:*

> *Section 4.1.3 is major in the discussion of significance test results. All the time lag plots*

*are discussed in Section 4.2. Figure 11 (e-f) are discussed in Section 4.2.3.*

p. 9, l. 32: Suggest to use "region of high significance" rather than "low fraction region" throughout the main text.

*Answer:*

*Ok, we changed seven places in the main text as follows.*

*Changes to paper:*

*(page 9, line 9-11) "As shown in the Figure,  fractions of less than 10 % (high significance) appear in the tropics for the altitude range of 40 – 60 km and 80 – 90 km, as well as at 40 °N for the altitude of about 65 km. The  high significance also appears at the high latitudes, ..."*

*(page 9, line 24-25) "The  regions of high significance obviously become smaller in Figure 9 (b–d),"*

*(page 10, line 6) "In total, the  region of high significance is larger for "summer" months (October – March) than "winter" months (April – September) for the global region."*

*(page 10, line 16) "The  region of high significance is larger for strong solar activity years than for weak solar activity years."*

*(page 10, line 17) "For weak solar activity years, the  region of high significance mainly concentrates in the equatorial mesopause region as shown in Figure 12 (b)."*

*(page 10, line 18) "For strong solar activity years, the  region of high significance is more distributed over high latitudes,"*

*(page 12, line 18) "Figure 15 (a) displays the sensitivity and shift of the  region of high significance for the latitude range from"*

p. 10, l. 16: "different input parameters". Better say "different settings", as input data (MLS, Mg II) remain the same.

*Answer:*

*Ok, changed.*

*Changes to paper:*

*(page 11, line 3-4) "Table 2 lists the sensitivity values (i.e., the slope of fitted linear regression line) and the uncertainties depending on the  different settings."*

p. 11, l. 10: "When comparing the graph with the significance test results shown in Figure 9 (a), it is apparent that the larger sensitivity values appear in regions with lower fraction, i.e., higher significance." If obvious, why mention it here (can be omitted).

*Answer:*

*We mentioned it here because we want to lead authors to compare the results of significance test and sensitivity analysis, i.e., Figure 9 (a) and Figure 13 (top panel).*

*Changes to paper:*

*(page 11, line 30-31) "When comparing the graph with the significance test results shown in Figure 9 (a), it  can be seen that the larger sensitivity values appear in*

*regions with lower fraction, i.e., higher significance, as expected."*

p. 11, l. 11: "determined time lag" –> "time lag"

*Answer:*

*Ok, done.*

*Changes to paper:*

*(page 11, line 32) "The bottom panel of Figure 13 shows the  time lag between local solar maximum (at the 27-day scale) and the temperature maximum."*

p. 11, l- l2: "Comparing the two panels of Figure 13 shows that large time lags tend to occur in latitude-altitude regions with small sensitivity". Why mention ii here (if greyed out in the plot because of no significance)

*Answer:*

*Yes, we are interested in the region of high significance, i.e., the region of high sensitivity in Figure 13 (top panel), so this sentence would be better if we changed as follows.*

*Changes to paper:*

*(page 11, line 32-33) "Comparing the two panels of Figure 13 shows that  small time lags tend to occur in latitude-altitude regions with large sensitivity."*

p. 12, l. 9: "obvious characteristics". What are they?

*Answer:*

*The obvious characteristics are the ones mentioned before regarding the sensitivity: the sensitivity increases in general with increasing latitude in the winter hemisphere and the sensitivity shows a tendency to increase with altitude in general in summer. To clarify this point, we changed the sentence as follows.*

*Changes to paper:*

*(page 12, line 29-30) "The shifts do not exhibit the same obvious latitude-altitude characteristics as the sensitivity, which is not further investigated here."*

p. 12, l. 27: "high significance fraction" –> "high significance region"

*Answer:*

*Ok, changed.*

*Changes to paper:*

*(page 13, line 17) "Similar to section 4.2.2, we investigate the sensitivity features for the high significance  region for different seasons as shown in Figure 15 (b – c)."*

Figure 6: Yellow curve is hard to see, suggest to plot first the unsmoothed curve and then overplot this with the smoother curves (order: red, blue, yellow). Then all curves may be visible. Yellow is hard to see, use another color for better legibility.

*Answer:*

*Ok, Figure 6 was changed. The text used to describe Figure 6 was rewritten in the manuscript as follows.*

**Changes to paper:**

(page 6, line 13) "The  red, blue and green points represent the local maxima identified for the 0-day, 7-day and 13-day smoothed Mg II index anomalies, respectively."

(page 24, Figure 6) "Figure 6. Top panel: Mg II index anomalies generated by subtracting a 35-day running mean from the time series. The black line presents the un-smoothed or 0-day smoothed Mg II index anomaly. The  red, blue and green points are the local maxima chosen from the 0-day, 7-day, and 13-day smoothed Mg II index anomalies, respectively. Bottom panel: similar to top panel except for the year 2005 only. In addition, the  red, blue and green lines present the Mg II index anomalies smoothed by a 0-day, 7-day and 13-day running mean, respectively."

*Fig. 6*

[Figure]

[Figure]

*to*

[Figure]

[Figure]

Figure 7: Show a vertical line in panel b, indicating the derived time lag.

*Answer:*

*Ok, changed as follows.*

*Changes to paper:*

*Fig.7(b)*
[Figure]
 *to*

*REFERENCE*

*Dudok de Wit, T., Kretzschmar, M., Lilensten, J., and Woods, T.: Finding the best proxies for the solar UV irradiance, Geophys. Res. Lett., 36, L10107, https://doi.org/10.1029/2009GL03782 5, 2009.*

*Snow, M., M. Weber, J. Machol, R. Viereck, and E. Richard, Comparison of Magnesium II core-to-wing ratio observations during solar minimum 23/24, J. Space Weather Space Clim., 4, A04, doi:10.1051/swsc/2014001, 2014.*

**Other revisions by authors**

*(1) Page 3, line 23*

    *Changes to paper:*

"To investigate  the robustness of the results, their dependence on parameters of the analysis methods (e.g., smoothing filter, window width and epoch centers),"

*(2) Page 4, line 13*

    *Changes to paper:*

"… Figure 1 shows the Mg II index data from 2005 to 2017  that is used in this analysis."

*(3) Page 5, line 32*

    *Changes to paper:*

"We define outliers as data points for which the  magnitude of the temperature anomaly exceeds 4 times the standard deviation of the anomaly time series."

*(4) Page 6, line 5*

    *Changes to paper:*

"Those steps above are a preparation for the  subsequent SEA,  significance testing and sensitivity analysis."

*(5) Page 6, line 27*

    *Changes to paper:*

"The epoch-averaged temperature anomaly also shows a clear maximum but with a time lag of 2 days, indicating that the response in mesospheric temperature to the solar forcing occurs with a  time lag."

*(6) Page 11, line 13*

    *Changes to paper:*

"Overall, there is a tendency  toward larger sensitivities if a wider window  is used for determining the anomalies."

*(7) Page 14, line 2*

    *Changes to paper:*

"For the analysis presented here it is important to remember that, for solar minimum conditions, the 27-day signatures are not statistically significant at most altitudes and latitudes."

*(8) Page 16, references*

    *Changes to paper:*

"Brasseur, G.  : The response of the middle atmosphere to longterm and short-term solar variability: A two dimensional model, J. Geophys. Res., 98, 23079 – 23090, https://doi.org/doi:10.1029/93JD02406, 1993."

*(9) Page 33, Figure 15*

*Changes to paper:*

*"Figure 15. Sensitivity in K (100 sfu) $^{-1}$ (red contour lines) and shift…"*